# B cell immunity to the Lassa virus glycoprotein is a correlate of vaccination-induced virus control in mice

Tiago Abreu-Mota ®[1,6] ✉, Anna-Friederike Marx[1,6], Dorothee Winterberg ®[2], Karen Tintignac[1], Jonas Fixemer[1], Florian Geier[1,3], Nicole Brodmann[1], Cemre Seven[1], Claudia Reichmuth ®[1], Anna Lena Kastner ®[1], Min Lu[1], Weldy V. Bonilla[1], Mirela Dimitrova[1], Mehmet Sahin[1], Gert Zimmer ®[4,5], Matthias Peipp[2] & Daniel D. Pinschewer ®[1] ✉

Lassa fever, a viral hemorrhagic fever caused by the arenavirus Lassa virus (LASV), affects thousands of individuals annually, highlighting the need for a vaccine. Yet, immunological correlates of viral load control remain poorly defined. Here we study vaccination-induced immunity in a surrogate LASV challenge model in mice. We find that LASV-cross-reactive B cell immunity induced by the glycoprotein of distantly related arenaviruses, such as lymphocytic choriomeningitis virus (LCMV), provides significant viral load control. Counter to common concepts, suppression of viremia is observed in the absence of CD8 T cells or neutralizing antibodies but correlates with non-neutralizing glycoprotein-specific antibody responses. Adoptive cell transfer experiments with monoclonal LCMV-specific B cells demonstrates that these cells suppress viral loads when previously activated by a heterologous cross-reactive glycoprotein and diversified by somatic hypermutation. These findings establish vaccination-induced B cell immunity to the LASV glycoprotein as a correlate of viral load control, independently of virus-neutralizing antibody titers at the time of challenge.

Lassa fever (LF), which is caused by infection with Lassa virus (LASV), is considered one of the most pressing health concerns in West Africa with no effective treatment or vaccine available for clinical use. Due to substantial habitat overlap between LASV's natural rodent reservoir and human settlements[1], zoonotic spillover is an ongoing reality. While historically estimated to account for ~100'000–300'000 human infections each year[2] more recent modeling of the virus' epidemiology projects 2.7 million annual LASV infections[3]. Although most LASV infections are thought to be subclinical, the LF fatality rate of hospitalized patients is in the range of 15%, and survivors often suffer from severe sequelae[4,5]. The considerable humanitarian and social burden caused by LF makes LASV a priority pathogen for both the World Health Organization (WHO) and for the Coalition for Epidemic Preparedness Innovations (CEPI). Indeed, the urgency for an effective vaccine is reflected by at least five registered vaccine candidates in clinical trials at the time of writing[6,7]. Nevertheless, the genetic diversity of LASV, evident in multiple lineages, still conveys doubt on whether a LF vaccine can be universally effective[8].

[1]Department of Biomedicine – Haus Petersplatz, Division of Experimental Virology, University of Basel, Basel, Switzerland. [2]Division of Antibody-Based Immunotherapy, Department of Medicine II, University Hospital Schleswig-Holstein, Kiel, Germany. [3]Swiss Institute of Bioinformatics, Department Biomedicine, University of Basel, Basel, Switzerland. [4]Institue of Virology and Immunology IVI, Mittelhäusern, Switzerland. [5]Department of Infectious Diseases and Pathobiology, Vetsuisse Faculty, University of Bern, Bern, Switzerland. [6]These authors contributed equally: Tiago Abreu-Mota, Anna-Friederike Marx. ✉e-mail: tiagojose.abreumota@unibas.ch; mottiago@gmail.com; Daniel.pinschewer@unibas.ch

LASV belongs to the Old World (OW) branch of the arenavirus family of enveloped negative-strand RNA viruses. Besides OW arenaviruses such as LASV, several New World (NW) arenaviruses including Machupo virus, Guanarito virus, and Junin Virus (JUNV) can cause hemorrhagic fever in humans[9]. For JUNV, the etiological agent of Argentine hemorrhagic fever (AHF), the vaccine Candid#1 is currently given to humans in endemic areas and provides disease protection[10]. Vaccination with Candid#1 or prior infection with JUNV induces neutralizing antibodies (nAbs) that serve as a correlate of protection[11]. In stark contrast, the presence of LASV-nAbs after infection is seldomly reported in humans and if so, these responses are generally delayed and low in magnitude[12–14]. This contrast has been attributed to a higher glycan density on the LASV envelope glycoprotein (GPC) than on JUNV GPC[15]. Consequentially, pre-clinical studies on a range of LF vaccine candidates have found that LASV-nAbs were inconsistently induced or reached only low titers[16–25], whereas robust T cell responses were commonly associated with vaccination-induced protection[22–25]. Moreover, studies comparing fatal and non-fatal LASV infection in unvaccinated non-human primates (NHPs) have observed that distinct innate immune response profiles as well as robust T cell responses predicted survival[26,27]. Based on the combined evidence, vaccine-mediated immunity to LF has been proposed to be predominantly T cell-based[28].

Despite generally low titers of LASV-nAbs in human survivors, the human immune system can form antiviral protective B cell clones. Originally established by anecdotal cases of successful convalescent serum therapy[14,29,30], the more recent isolation of a panel of monoclonal LASV-nAbs[31,32] has allowed for a more systematic assessment of both the binding behavior and the protective efficacy of LASV-nAbs[33]. Experimental work in guinea pigs and NHPs found that monoclonal antibody (mAb) therapy suppressed viremia within forty-eight hours and prevented disease even when therapeutically administered eight days after LASV exposure[34–37]. Consistent with this, several large-scale clinical studies conducted in both children and adults, across different countries and over multiple decades, have identified viral load, measured either on hospital admission or during the course of illness, as the primary predictor of LF outcome[38–41], suggesting that viral-load suppression should be a central goal of vaccination-induced immunity. While neutralization is traditionally considered the best mechanistic correlate of antibody-mediated protection against viral disease[42], growing evidence emphasizes an important additional role for vaccination-induced non-neutralizing antibodies (non-nAbs) in preventing LF[17–20]. Thereby, LASV follows the example of many other viruses such as HIV[43,44], Ebola virus[45], influenza virus[46,47], vaccinia virus[48] and lymphocytic choriomeningitis virus (LCMV)[49–52] for which the same has been shown beforehand.

Several independent studies suggest that exposure to a heterologous LASV lineage, or even to more distantly related OW arenaviruses such as Mopeia (MOPV) or LCMV can generate immunity to LASV in guinea pigs as well as NHPs[53,54]. This was commonly accredited to T cell epitope conservation between LASV and other OW arenaviruses. Passive transfer studies in guinea pigs revealed that splenocytes from OW arenavirus-immunized animals protected against lethal LASV challenge while passive serum transfer did not. This was interpreted to reflect T cell- rather than B cell-mediated cross-protection[54,55]. However, these studies did not consider the possibility that recall responses by virus-primed memory B cells, contained in adoptively transferred splenocyte preparations, could have contributed to the prevention of fatal disease.

In the present study we show that vaccination using the GPC of distantly related OW arenaviruses induces LASV-GPC-specific B cell responses that suppress viral loads independently of memory CD8 T cells. The identification of secondary B cell responses to LASV-GPC as a correlate of viral load control has conceptual and practical implications for vaccine design and testing.

## Results

### The glycoproteins of LASV and LCMV elicit cross-reactive antibody responses

The bi-segmented arenavirus genome consists of a large (L) and a small (S) RNA segment. The latter encodes the glycoprotein complex (GPC; Fig. 1D), which is post-translationally cleaved into GP1 and GP2 subunits. It decorates infectious virions and mediates receptor binding as well as membrane fusion, and therefore represents the only target of virus-nAbs. Sequence conservation between the GPCs of different LASV lineages can be as low as ~93% amino acid identity (Fig. 1A, B). When $LASV_{LIV}$-GPC was compared to GPCs of other OW arenaviruses such as Merino-Walk virus (MWV) and LCMV these values dropped to ~60-70% but were still higher when compared to a NW arenavirus such as Pichinde (PICV; ~40%). Despite the sequence divergence between different OW arenaviruses, the monoclonal neutralizing antibodies (mAbs) 12.1 F and 18.5C[31], isolated from LF convalescent patients, bound to the GPCs of both LCMV and LASV (Fig. 1C). Likewise, the mAbs KL25[56] and WEN3[57], which were isolated from LCMV-infected mice, exhibited $LASV_{LIV}$-GPC-specific reactivity, albeit WEN3 binding was comparably weak. These observations indicated that both mouse and human B cell repertoires can give rise to antibodies that cross-react between different OW arenavirus GPCs. LASV requires biosafety level 4 (BSL-4) containment, complicating experimental work with this pathogen. To investigate the induction of OW arenavirus GPC-cross-reactive antibodies in a small animal model under BSL-2 conditions we turned to a recombinantly engineered LCMV expressing LASV-GPC instead of LCMV-GPC (Fig. 1D, E). This virus does not establish viremic infection in mice[58,59] and hence is referred to as "acute variant" (rLCMV/ $LASV_{L(x)}^A$). Here we found that viremia was not detected even when mice were depleted of CD8 T cells (Fig. 1F, S2A, B). Instead, achieving protracted viremia akin to rLCMV infection required the use of an rLCMV/LASV virus expressing a chimeric GPC consisting of the extracellular and transmembrane domains of LASV-GPC fused to the cytoplasmic domain of LCMV-GPC (Fig. 1E, G; termed rLCMV/$LASV_{L(x)}^C$, "chronic variants"). Mice infected with these "chronic variants" developed peak viremia of about $10^5$ focus-forming units per milliliter of blood, comparable to LASV loads documented in humans, guinea pigs, and select NHP species during fatal LF[21,41,55,60,61]. The importance of the LCMV-GPC cytoplasmic tail for the replication of rLCMV/LASV chimeric viruses in mice had previously been assessed for viruses expressing the LASV lineage IV GPC (rLCMV/$LASV_{LIV}^A$, rLCMV/ $LASV_{LIV}^C$)[58]. Here we extended and generalized these findings by testing viruses expressing not only the LASV lineage II GPC (Fig. 1E–G; rLCMV/$LASV_{LII}^A$, rLCMV/$LASV_{LII}^C$) but also the ones of LASV LI and LIII GPCs (Fig. S1A, B). Conversely, rLCMV expressing a chimeric rLCMV-GPC ectodomain fused to the $LASV_{LIV}$-GPC cytosolic tail (rLCMV[A]) failed to establish detectable viremia, further tying the GPC C-terminal cytosolic tail to LCMV fitness in mice (Fig. 1F, compare to Fig. 1G). Sera collected on day 35 after rLCMV infection contained IgG that bound not only to LCMV-GPC but also cross-reacted to $LASV_{LIV}$-GPC (Fig. 1H, I), albeit to a lower extent. Correspondingly, serum IgG from rLCMV/ $LASV_{LI}^C$- rLCMV/$LASV_{LII}^C$-, rLCMV/$LASV_{LIII}^C$-and rLCMV/$LASV_{LIV}^C$-infected mice bound not only to $LASV$-$GPC_{LIV}$ but also cross-reacted to the LCMV-GPC (Fig. 1H–I, S1C, D). Taken together, we observed substantial serological cross-reactivity between the GPCs of all four LASV lineages and consistent cross-reactivity to the GPC of the distantly related LCMV.

### Antibody cross-reactivity between LASV GPCs of different lineages suppresses viremia

Next, we tested whether the observed serological cross-reactivity between different LASV lineage GPCs afforded viral load control. We immunized mice with the aviremic rLCMV/$LASV_{LII}^A$ variant and challenged them four weeks later with a virus expressing a heterologous LASV-GPC, rLCMV/$LASV_{LIV}^C$ (Fig. 2A). rLCMV/$LASV_{LII}^A$ rather than its

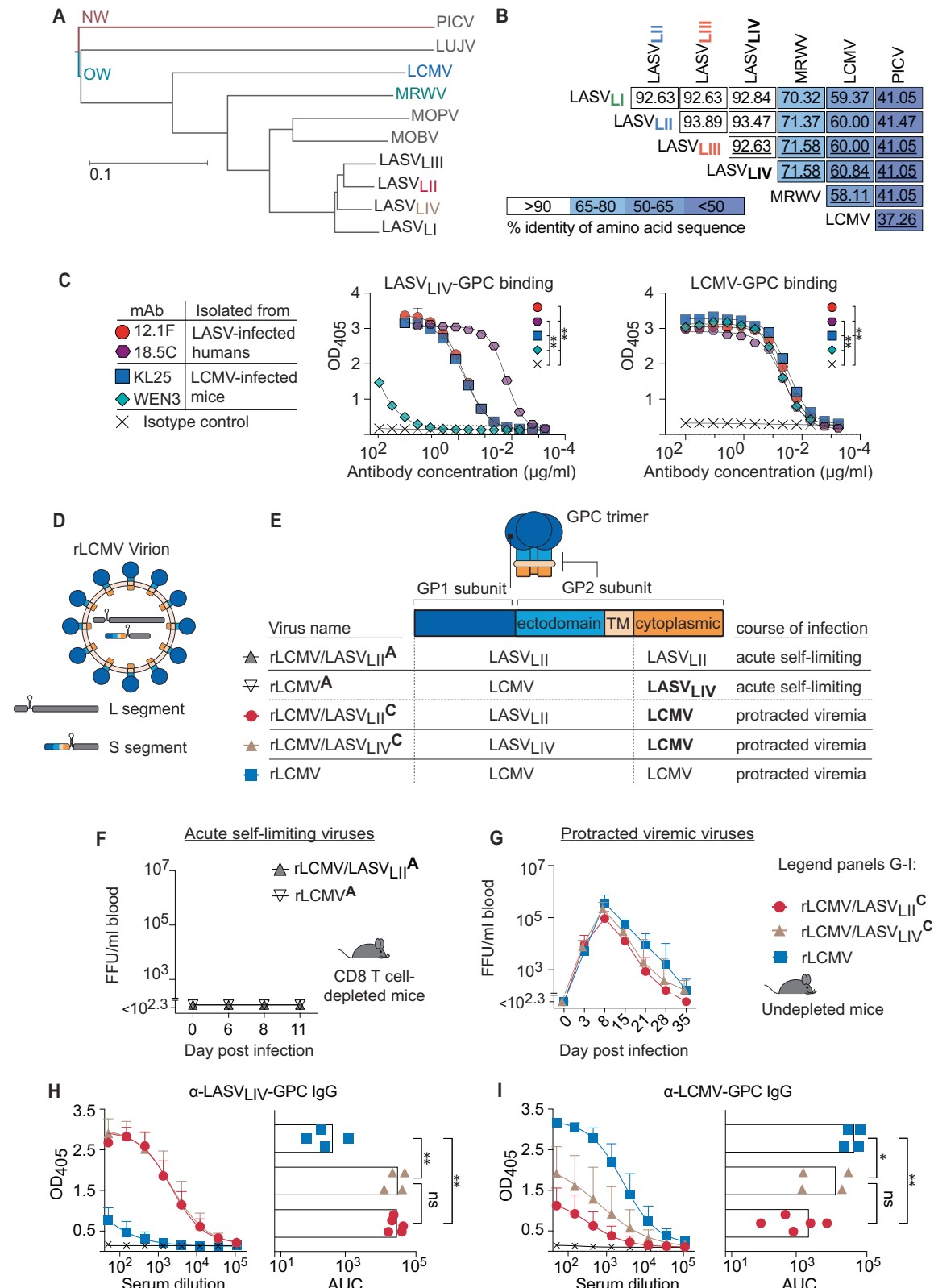

chronic counterpart rLCMV/LASV_LII^C was used for immunization in order to prime the immune system with LASV_LII-GPC without establishing viremic infection that would have interfered with the determination of rLCMV/LASV_LIV^c viremia upon challenge. To control for the immunological specificity of vaccination-induced viral load suppression, a separate group of mice was immunized with an engineered LCMV (rLCMV/VSVG), which is molecularly identical to rLCMV/LASV_LII^A

but expresses the antigenically unrelated vesicular stomatitis virus glycoprotein (VSVG) instead of LASV-GPC. Additional control animals were left unimmunized (none; Fig. 2A). To selectively assess viral load control by cross-reactive antibody responses without a contribution from vaccination-induced CD8 T cells, the latter were depleted prior to immunization and again prior to challenge (Fig. 2A, S2A, B). rLCMV/LASV_LII^A but not rLCMV/VSVG induced LASV_LIV-GPC-specific IgG, as

**Fig. 1 | The glycoproteins of LASV and LCMV elicit cross-reactive antibody responses. A** Phylogenetic tree depicting GPC amino acid distance between selected OW arenaviruses and Pichinde virus (PICV) as a New World (NW) arenavirus representative. **B** Percentage of amino acid sequence identity between the GPC of members of the four Lassa virus (LASV) lineages, Merino Walk virus (MRWV), lymphocytic choriomeningitis virus (LCMV) and PICV. **C** Binding of the indicated LASV- and LCMV-induced mAbs to $LASV_{LIV}$-GPC and to LCMV-GPC in ELISA. Symbols represent the mean ± SEM of three technical replicates. **D** Schematic of an LCMV virion with its GPC (for simplicity represented as a monomeric GP1-GP2 heterodimer) and the topology of each GPC domain. The viral genomic large (L) and small (S) segments are schematically depicted inside the virion. **E** Table listing the recombinant LCMV viruses used in this study, the origin of each GPC domain they express and the course of infection they elicit in mice. **F** WT

mice depleted of CD8 T cells or left undepleted (**G**) were infected with the designated viruses. Viremia was measured over time. **H, I** Serum IgG titers against $LASV_{LIV}$-GPC (**H**) and against LCMV-GPC (**I**) at d35 of the experiment in (**G**). ELISA curves and their area under the curve (AUC) are shown. **C** Statistical significance between the AUC values from the respective mAbs was determined by 1-way ANOVA with Tukey's post-hoc analysis performed for pairwise comparisons. Symbols in (**F, G**) and in ELISA curves of (**H, I**) represent the mean+SD of n = 5 (rLCMV/$LASV_{LII}$^C) and n = 4 mice (rLCMV/$LASV_{LIV}$^C and rLCMV groups) Symbols on AUC bars of (**H, I**) represent the same individual mice. One representative experiment out of two similar ones is shown (**C, F–I**). *$p \leq 0.05$; **$p \leq 0.01$; n.s.: $p > 0.05$ by 1-way ANOVA with Bonferroni's post-test (**C, H, I**). In (**C**) only significant differences are shown.

expected (Fig. 2B; compare Fig. 1H). Upon rLCMV/$LASV_{LIV}$^C challenge both rLCMV/VSVG-immunized animals and unimmunized controls developed high-level viremia throughout the observation period of three weeks whereas rLCMV/$LASV_{LII}$^A-immune mice exhibited low to undetectable viral loads (Fig. 2C). Suppression of viremia by rLCMV/$LASV_{LII}$^A-immune but not by rLCMV/VSVG-vaccinated animals indicated that rLCMV/$LASV_{LIV}$^C control relied largely on specific immunity to LASV-GPC. At the end of the experiment, the rLCMV/$LASV_{LII}$^A-immune aviremic group harbored significantly higher levels of anti-$LASV_{LIV}$-GPC-specific antibodies than either of the control groups (Fig. 2D). While $LASV_{LIV}$-neutralizing antibodies (nAb) were undetectable in all groups (Fig. 2E), a surrogate assay for antibody-dependent cell-mediated cytotoxicity (ADCC) demonstrated that serum antibodies from rLCMV/$LASV_{LII}$^A-immune mice triggered Fc gamma receptor IV- (FcγRIV-) mediated effector cell activation in the presence of target cells expressing LASV-GPC (Figure S2C). To determine whether rLCMV/$LASV_{LII}$^A-induced immunity against rLCMV/$LASV_{LIV}$^C was antibody-dependent we exploited sIgM⁻/⁻xAID⁻/⁻ mice, whose B cells are unable to secrete immunoglobulin[62]. When tested in an experimental set-up as outlined above (Fig. 2A) immunized sIgM⁻/⁻xAID⁻/⁻ mice failed to mount $LASV_{LIV}$-GPC-specific antibodies, as expected, and exhibited similar viral loads as unimmunized WT controls (Fig. 2F, G). Overall, these data established LASV lineage cross-reactive humoral immune control of high-level viremia independently of neutralizing serum activity. Of note, however, rLCMV/$LASV_{LII}$^A immunization elicited also viral epitope-specific CD8 T cell responses (Figure S2D), and rLCMV/$LASV_{LII}$^A-immunized sIgM⁻/⁻xAID⁻/⁻ mice that were not depleted of CD8 T cells completely suppressed rLCMV/$LASV_{LIV}$^C challenge viremia (Figure S2E). This observation indicated that not only specific antibodies but independently also vaccination-induced CD8 T cells were able to contain rLCMV/$LASV_{LIV}$^C challenge viremia, such that assessing the antiviral efficacy of the former required depletion of the latter.

Given that LCMV-immune serum cross-reacted to LASV-GPCs (Fig. 1H), we tested whether LCMV infection can induce LASV-GPC-specific antibody immunity to suppress viremia. To this end, we used LCMV Armstrong (LCMV-Arm), a viral strain that elicits only acute infection in mice and is effectively controlled by CD8 T cells[63–65] (Fig. 2H). Additional mice were immunized with either rLCMV/$LASV_{LII}$ as positive control, or with rLCMV/VSVG, as negative control. Sera collected 4 weeks post-infection revealed that LCMV-Arm infection induced $LASV_{LIV}$-GPC-specific IgG titers that were lower than those of rLCMV/$LASV_{LII}$-immune sera but significantly above rLCMV/VSVG-immune background (Fig. 2I). As expected, LCMV infected mice had higher LCMV-GPC-specific IgG titers than either control group (Figure S2F, G). To determine whether LCMV-Arm-induced B cell immunity afforded LASV-GPC-specific viral load control, all groups were CD8 T cell-depleted and challenged with rLCMV/$LASV_{LIV}$^C. LCMV-Arm-immunized animals developed transient low-level viremia, which subsided three weeks post-challenge (Fig. 2J). rLCMV/$LASV_{LII}$^A-immune mice remained aviremic at all time points whereas rLCMV/VSVG-

immunized mice exhibited persistent viremia throughout the experiment (Fig. 2J). Within three weeks after challenge, $LASV_{LIV}$-GPC-specific IgG titers of LCMV-Arm-immunized mice increased substantially, reaching levels comparable to those of the LCMV/$LASV_{LII}$-immunized group and higher than in rLCMV/VSVG-immunized animals or in unimmunized controls (Fig. 2K, L). LCMV-GPC-specific IgG titers did not increase after challenge of LCMV-Arm-immunized mice and increased modestly in rLCMV/$LASV_{LII}$-immune animals (S2H, I). Importantly, none of the groups exhibited LASV-nAb titers (Fig. 2M). Collectively, these data show that pre-existing GPC-cross-reactive B cell responses can mediate viral load control.

## LCMV-GP1-specific B cells proliferate and affinity mature in response to LASV-GPC

The rapid rise of LASV-GPC-specific antibody titers and suppression of viremia in LCMV-Arm-immune and rLCMV/$LASV_{LIV}$^C-challenged animals (Fig. 2L) suggested a recall response by LASV – LCMV-GPC cross-reactive memory B cells. We hypothesized that LCMV-induced B cells with low-level cross-reactivity could undergo additional affinity maturation upon recall by LASV-GPC. Hence, we assessed whether monoclonal B cells from HkiL mice, engineered to express the LCMV-GP1-specific mAb KL25 as B cell receptor (BCR)[66], can react to LASV-GPC. The KL25 mAb bound to the GPCs of $LASV_{LIV}$ and $LASV_{LII}$, albeit less potently than to LCMV-GP1, and it neutralized LCMV but not rLCMV/$LASV_{LIV}$^C in cell culture (Fig.1C, Fig. 3A, B). Accordingly, KL25 suppresses LCMV viremia when passively administered to mice[67]. Here we infected mice with rLCMV/$LASV_{LIV}$^C, and treated them with passive antibody one day later. To determine direct antiviral effects we assessed viremia on day 5 after infection i.e. during the antibody's first half-life period[68] and prior to the onset of the CD8 T cell response[69]. KL25 failed to measurably reduce viral loads in rLCMV/$LASV_{LIV}$^C-infected mice, whereas mice treated with the LASV-GP1-specific mAb 12.1 F were aviremic (Fig. 3C). Next, we labeled KL25 BCR-expressing B cells of HkiL mice ("HkiL cells"; CD45.1⁺) with CFSE and adoptively transferred them into mice infected with rLCMV expressing either one of a range of different glycoproteins (Fig. 3D, Figure S3B). As evident from the dilution of CFSE and from the number of cells recovered five days later, HkiL cells proliferated in response to rLCMV expressing the GPCs of $LASV_{LI}$, $LASV_{LII}$, $LASV_{LIII}$ and $LASV_{LIV}$ but not to rLCMV/VSVG in here serving as negative control (Fig. 3E, F, Figure S3A–D). When more broadly assessing reactivity to other OW arenaviruses (Fig. 1A) we found that the GPC of Merino Walk virus (MRWV) but not those of Mobala or Lujo virus triggered HkiL cell proliferation (Fig. 3E, F, S3C, D). Next, we tested whether HkiL cells can affinity mature to increase their LASV-GPC binding capacity. HkiL cells were transferred to BCR-restricted mice (TgL), infected with either rLCMV/$LASV_{LIV}$^C or rLCMV (Fig. 3G). Five weeks after infection, $LASV_{LIV}$-GP1-specific IgG titers were similar between rLCMV/$LASV_{LIV}$^C- and LCMV-infected recipients of HkiL cells (Fig. 3H). These antibodies were largely produced by transferred HkiL cells, since titers in rLCMV/$LASV_{LIV}$^C -infected TgL controls

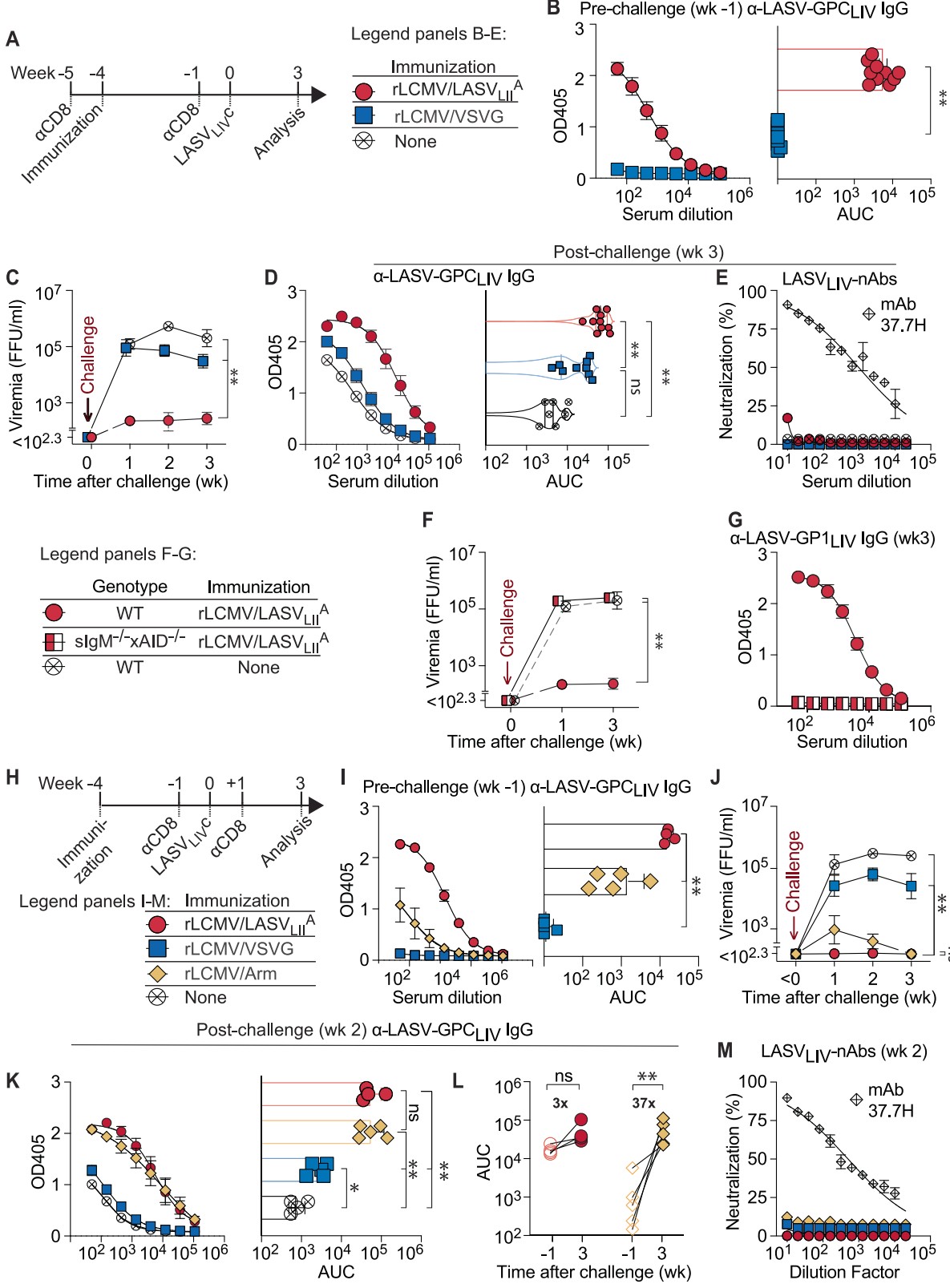

without HkiL cell transfer were undetectable (Figure S3E, F). In keeping with the ELISA results (Fig.3H), LCMV-nAb titers of the two groups of HkiL cell recipients were in similar ranges (Fig. 3I), confirming that rLCMV/LASV$_{LIV}^C$ and LCMV triggered an HkiL cell response of comparable magnitude. However, rLCMV/LASV$_{LIV}^C$-infected but not rLCMV-infected HkiL cell recipients exhibited substantial rLCMV/LASV$_{LIV}^C$-nAb titers (Fig. 3J, K), suggesting that HkiL

cells responding to rLCMV/LASV$_{LIV}^C$ underwent affinity maturation that resulted in improved LASV-GPC binding.

## LASV-GPC but not LCMV-GPC priming enables control of rLCMV/LASV$_{LIV}^C$ viremia by HkiL cells

The results presented above validated the HkiL cell transfer system to investigate viral load control by OW arenavirus cross-reactive B cells.

**Fig. 2 | Antibody cross-reactivity between LASV GPCs of different lineages suppresses viremia. A** Experimental outline for panels (**B**–**E**). WT mice were depleted of CD8 T cells at wk −5, immunized with rLCMV/LASV$_{LII}$$^A$ or rLCMV/VSVG at wk −4 or left unimmunized, again depleted of CD8 T cells at wk −1 and challenge with rLCMV/LASV$_{LIV}$$^C$ at wk 0. **B** Pre-challenge (wk-1) anti-LASV$_{LIV}$-GPC IgG titers. **C** rLCMV/LASV$_{LIV}$$^C$ viremia over time (**D, E**). Anti-LASV$_{LIV}$-GPC IgG and LASV$_{LIV}$-nAbs at wk 3. mAb 37.7H served as positive control (PC) in (**E**). Results in (**A**–**E**) show combined data from two independent experiments with $n = 7$ (none), $n = 9$ (rLCMV/ VSVG) and $n = 10$ (rLCMV/LASV$_{LII}$), mice and are expressed as mean ± SD, while symbols reporting AUC show the same individual mice. **F, G** WT and sIgM$^{−/−}$xAID$^{−/−}$ were subject to an experiment analogous to (**A**). **F** rLCMV/LASV$_{LIV}$$^C$ viremia over time from two combined experiments. **G** Anti-LASV$_{LIV}$-GPC IgG titers at wk 3. The mean ± SD of $n = 11$(rLCMV/LASV$_{LII}$), $n = 10$ (sIgM$^{−/−}$xAID$^{−/−}$) and $n = 5$ (none) mice

from two independent experiments is shown. **H** Experimental outline for panels (**I**−**M**). WT mice were immunized with the indicated viruses at wk −4, depleted of CD8 T cells at wk −1, challenged with rLCMV/LASV$_{LIV}$$^C$ at wk 0, and depletion of CD8 T cells was repeated at wk +1. **I** Anti-LASV$_{LIV}$-GPC IgG titers at wk −1. **J** rLCMV/ LASV$_{LIV}$$^C$ viremia over time (**K**). Anti-LASV$_{LIV}$-GPC IgG and (**L**) matched proportional increase of anti-LASV$_{LIV}$-GPC IgG AUC from wk −1 to wk 3. **M** LASV$_{LIV}$-nAbs at wk 3 with mAb 37.7H as positive control. In (**I**−**M**) the mean ± SD of $n = 5$ (rLCMV/Arm), $n = 4$ (rLCMV/LASV$_{LII}$) and $n = 5$ (rLCMV/VSVG) mice from one out of two experiments is shown. Symbols reporting AUC show the same individual mice. Two-way ANOVA with Sidak's post-test was used in (**C, F, J**) to compare rLCMV/LASV$_{LII}$ -vaccinated mice against all other groups, and in (**L**) to assess differences between pre- and post-challenge; One-way ANOVA with Tukey's post-test was performed in (**D, I, K**), and two-tailed Welch's $t$-test in (**B**). *$p ≤ 0.05$; **$p ≤ 0.01$; ns: $p > 0.05$.

Next, we sought to understand whether previously activated HkiL cells, unlike their naïve counterpart, could suppress rLCMV/LASV$_{LIV}$$^C$ viremia. To that end, we transferred HkiL cells into TgL recipients and immunized them with rLCMV/LASV$_{LII}$$^A$ ("HkiL+LASV$_{LII}$$^{A*}$", Fig. 4A). Control groups consisted of TgL mice that were immunized with rLCMV/ LASVL$_{LII}$$^A$ but did not receive HkiL cells ("LASV$_{LII}$$^A$-only") and of TgL mice that received HkiL cells but were not immunized ("HkiL-only"). Four weeks later all mice were challenged with rLCMV/LASV$_{LIV}$$^C$. Importantly, all groups were depleted of CD8 T cells prior to immunization and again prior to challenge. Animals in the HkiL+LASV$_{LII}$$^A$ group developed only low to undetectable levels of rLCMV/LASV$_{LIV}$$^C$ viremia, whereas substantial viremia was measured in the LASV$_{LII}$$^A$-only and the HkiL-only control groups (Fig. 4B). Of note, this experiment and similar ones below were ended at around two weeks after challenge since persisting low-level rLCMV/LASV$_{LIV}$$^C$ viremia in a subset of HkiL+LASV$_{LII}$$^A$ animals was associated with mutations at amino acid 114 of the viral GPC, which corresponded to known KL25 escape mutations in LCMV-GPC[70] (Figure S4A, B). The absence of such mutations from viruses circulating in LASV$_{LII}$$^A$-only mice indicated these mutations were the result of HkiL cell-driven in vivo selection, and mutational escape would, therefore, have precluded a meaningful interpretation of the association between HkiL cell responses and viral load control at later time points. Two weeks after rLCMV/LASV$_{LIV}$$^C$ challenge LCMV-nAb titers were negligible in LASV$_{LII}$$^A$-only mice but high in the HkiL+LASV$_{LII}$$^A$ and HkiL-only groups, indicating that the transferred HkiL cells were the source of the antibody response in the latter groups (Fig. 4D). However, the LASV$_{LIV}$-nAb titers of HkiL+LASV$_{LII}$$^A$ mice were substantially higher than those of HkiL-only animals. Mice in the LASV$_{LII}$$^A$-only group had no detectable LASV$_{LIV}$-nAb titers (Fig. 4C). When standardizing LASV$_{LIV}$-nAb titers to LCMV nAb titers (LASV$_{LIV}$: LCMV) the HkiL+LASV$_{LII}$$^A$ group had an ~30-fold higher response (Fig. 4E) than the HkiL-only group. The ADCC surrogate assay revealed further that the serum of HkiL+LASV$_{LII}$$^A$ mice triggered substantially higher FcγRIV-mediated effector cell activation in the presence of LASV-GPC-expressing target cells than sera of the HkiL-only control group (Figure S4C). These findings suggested that antibodies produced by naïve HkiL cells did not neutralize LASV-GPC and failed to mediate substantial FcγR-mediated effector cell activation or to suppress rLCMV/LASV$_{LIV}$$^C$ viremia, (concurrent with Fig. 3C), whereas HkiL cells of the HkiL+LASV$_{LII}$$^A$ group had become antivirally effective by the time of challenge.

Viral load control in the HkiL+LASV$_{LII}$$^A$ group could have been due to HkiL cell affinity maturation upon rLCMV/LASV$_{LII}$$^A$ immunization or might have simply reflected an accelerated secondary response of previously activated HkiL cells. In an experiment analogous to the above (Fig. 4F), HkiL cells were adoptively transferred to TgL mice that were immunized with either rLCMV/LASV$_{LII}$$^A$ (HkiL+LASV$_{LII}$) or rLCMV$^A$ (HkiL+LCMV) and challenged with rLCMV/LASV$_{LIV}$$^C$ four weeks later. As observed above, mice of the HkiL+LASV$_{LII}$ group developed only low to undetectable levels of viremia while both the HkiL+LCMV group as well

as the LASV$_{LII}$$^A$-only control group were viremic (Fig. 4G). One week prior to challenge, the HkiL+LCMV group exhibited higher LCMV-GP1- and LASV-GP1-binding IgG titers than the HkiL+LASV$_{LII}$ group (Figure S4E) while neither group had LASV$_{LIV}$-nAb titers (Figure S4D). Within two weeks after challenge, however, LASV-GP1-binding IgG increased significantly in the HkiL+LASV$_{LII}$ group but not in the HkiL+LCMV group (S4F). Moreover, when assessed by competition ELISA, sera from the HkiL+LASV$_{LII}$ group bound better to KL25-blocked LASV-GP1 than sera from HkiL+LCMV mice indicating affinity maturation of LASV$_{LII}$ -primed HkiL cells (Figure S4G). In line with these findings, mice in the HkiL+LASV$_{LII}$ group had higher LASV$_{LIV}$-nAb titers as well as a higher LASV: LCMV nAb ratio than HkiL+LCMV animals (Fig. 4H–J).

Overall, the superior antiviral activity of rLCMV/LASV$_{LII}$$^A$- than of rLCMV$^A$-primed HkiL cells and a correspondingly improved antibody response to LASV supported the hypothesis that affinity maturation in response to LASV-GPC$_{LII}$ rendered HkiL cells effective against rLCMV/ LASV$_{LIV}$$^C$.

In a separate experiment we enumerated and characterized HkiL cells of HkiL+LASV$_{LII}$, HkiL+LCMV and HkiL-only animals two weeks after rLCMV/LASV$_{LIV}$$^C$ challenge (Fig. 4K–M). HkiL+LASV$_{LII}$ mice had fewer CD138$^+$CD19$^-$ antibody-secreting cells (ASCs) and also fewer CD38$^-$Bcl6$^+$ germinal center (GC) B cells than the other two groups (Fig. 4K–M). Attenuated HkiL cell expansion in HkiL+LASV$_{LII}$ mice was likely due to the cells' reduced triggering under conditions of efficient viral load control. In contrast, HkiL memory B cells (MBCs; CD38$^+$Bcl6$^-$) of HkiL+LASV$_{LII}$ mice were in similar ranges as in HkiL+LCMV controls and more abundant than in the HkiL-only group, resulting in a proportional overrepresentation of MBCs in the HkiL compartment of HkiL+LASV$_{LII}$ mice (Fig. 4L). These data indicated that the failure of HkiL+LCMV mice to control rLCMV/LASV$_{LIV}$$^C$ was not due to inefficient expansion of LCMV-primed HkiL cells upon challenge but rather suggested that rLCMV/LASV$_{LIV}$$^C$ control required HkiL cells to undergo affinity maturation.

## Suppression of rLCMV/LASV$_{LIV}$$^C$ viremia requires HkiL cell receptor diversification by hypermutation

To compare the ability of different viral glycoproteins to promote HkiL cell receptor hypermutation and clonal diversification, we vaccinated TgL recipients of HkiL cells with either rLCMV/LASV$_{LII}$$^A$ or rLCMV$^A$ and five weeks later we processed progeny HkiL GC B cells and MBCs for single cell RNA sequencing-based V(D)J determination (Fig. 5A, S5A). At the time of analysis the repartition of HkiL progeny into GC B cells, MBC, and ASCs was comparable irrespective of the vaccine administered (Fig. 5B). In notable contrast, the antibody light and heavy chain complementarity-determining regions (CDRs) of rLCMV/LASV$_{LII}$$^A$-induced HkiL GC B cells and also the light chain CDRs of MBCs from the same group of mice exhibited a significantly higher mutational burden than the respective sequences of rLCMV$^A$-activated HkiL cells (Fig. 5C–F). Accordingly, the clonal diversity of HkiL GC B cells was

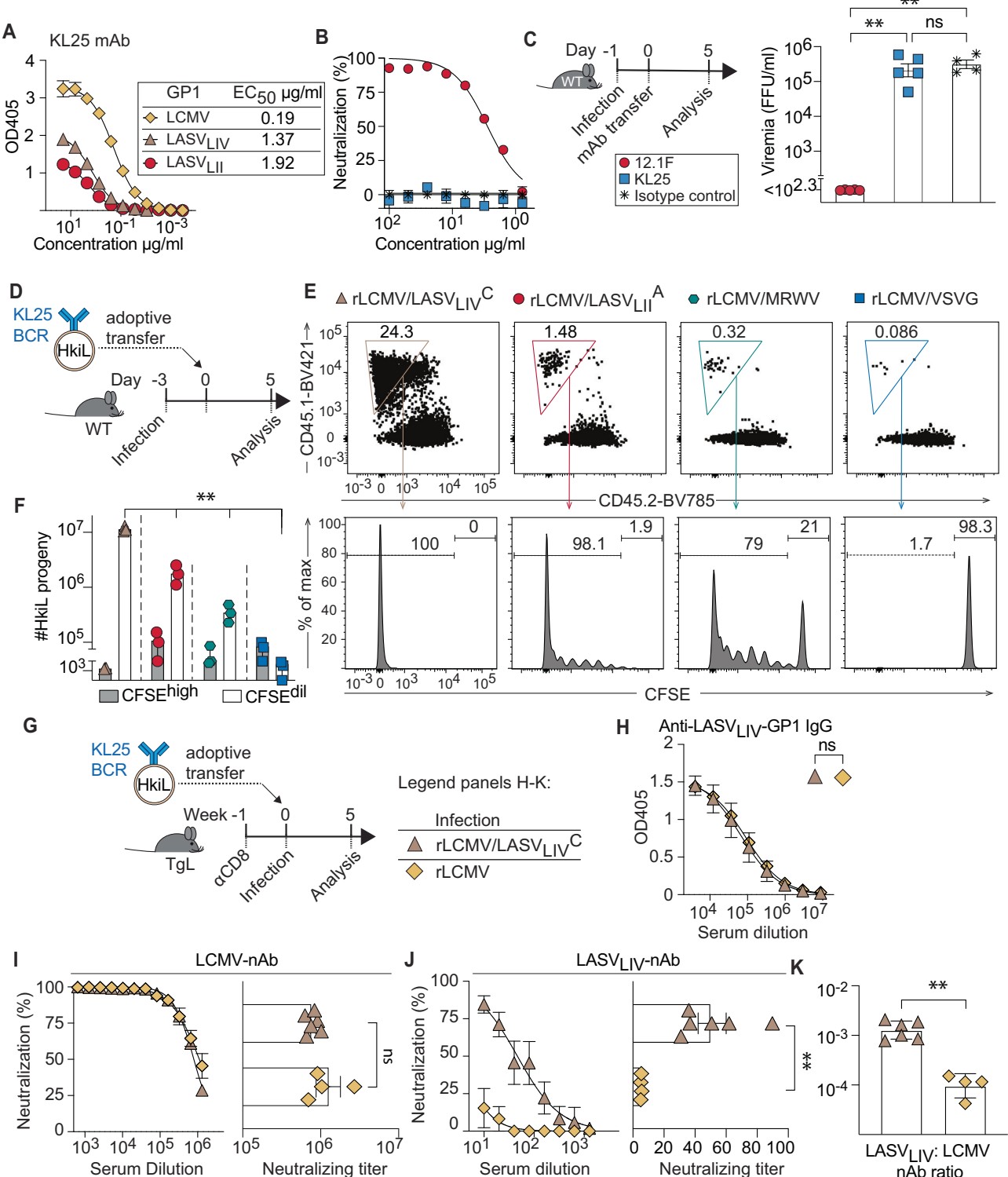

higher in rLCMV/LASV_LII^A- than in rLCMV^A-vaccinated animals as also evident in a higher Gini-Simpson diversity index, and an analogous trend was noted for HkiL MBCs (Fig. 5G, H). The observed CDR hypermutation exhibited a clear pattern with prominent recurrent amino acid exchanges (Figure S5B, C).

To test whether hypermutation was rate-limiting for the antiviral effects of rLCMV/LASV_LII^A-primed HkiL cells we used HkiL^AIDG23S mice. These mice carry a point mutation (AIDG23S) in the *aicda* gene encoding for activation-induced cytidine deaminase (AID), which is reported to reduce somatic hypermutation rates by up to -10-fold[71],

thus curbing BCR diversification in germinal centers. CD8 T cell-depleted TgL recipients of HkiL or HkiL^AIDG23S B cells were immunized with either rLCMV/LASV_LII or rLCMV^A and challenged with rLCMV/LASV_LIV^C four weeks later (Fig. 6A). Consistent with Fig. 4, the HkiL+LASV_LII group controlled viral loads whereas HkiL+LCMV and HkiL^AIDG23S + LCMV mice developed substantial viremia (Fig. 6B). Importantly, the HkiL^AIDG23S + LASV_LII group only transiently suppressed rLCMV/LASV_LIV^C, with increased viral loads on day 14, demonstrating that HkiL cell hypermutation was rate-limiting for their efficacy against rLCMV/LASV_LIV^C viremia. Two weeks post- challenge, sera from

**Fig. 3 | LCMV-GP1-specific B cells proliferate and affinity mature in response to LASV-GPC. A** Binding of the LCMV-neutralizing mAb KL25 to the GP1 of LCMV, LASV_LIV and LASV_LII in ELISA. **B** LASV_LIV-neutralizing activity of the mAbs KL25 and 12.1 F. Symbols in (**A**, **B**) represent the mean of two technical replicates. **C** Assessment of the ability of the KL25 mAb to suppress rLCMV/LASV_LIV^C viremia. WT mice were infected with rLCMV/LASV_LIV^C on d-1, and injected with 1 mg of KL25 ($n = 5$), 12.1 F ($n = 5$), or isotype control ($n = 4$). Viremia was determined at d5. **D** WT mice were infected at d-3 with either rLCMV/LASV_LIV^C, rLCMV/LASV_LII^A, rLCMV/ MRWV or rLCMV/VSVG at d-3, received 10^6 CFSE-labeled HkiL splenocytes by adoptive transfer at d0, followed by flow cytometric analysis on d5 (**E**). Gating of adoptively transferred HkiL cells in spleen (CD45.1^+, top row) and determination of CFSE dilution (bottom row). Representative ($n = 3$ per group) FACS plots are shown, numbers indicate the percentage of gated cells. **F** Total splenic count of non- proliferated (CFSE^high) HkiL cells and of proliferated HkiL cells with diluted CFSE levels (CFSE^dil) from the experiment in (**E**). **G, K** TgL mice were depleted of CD8 cells at week −1, infected with either rLCMV ($n = 4$) or rLCMV/LASV_LIV^C ($n = 5$) at day −1 (TgL) and given 10^4 HkiL splenocytes by adoptive transfer, followed by analysis at wk 5. **H** Anti-LASV_LIV-GP1 IgG by ELISA. **I, J** LASV_LIV-nAb and LCMV-nAb responses with IC_{50} titers. **K** Ratio of the LASV_LIV to LCMV nAb IC_{50} titers. Symbols in (**C, F, K** and right hand graphs of **I, J**) represent individual mice, bars in (**C, F, I, J, K**) show the mean ± SD. Symbols in (**H** and left hand graphs of **I, J**) represent the mean ± SD. Data in (**A**–**K**) are from one out of two similar experiments. One-way ANOVA with Tukey's multiple comparison post-hoc analysis (**C**), one-way ANOVA with Dunnett's multiple comparison post-hoc analysis comparing rLCMV/VSVG- immunized mice against all other groups (**F**) and two-tailed Welch's *t*-test (H,I,J,K) were performed for statistical analysis. **p ≤ 0.01; ns: p > 0.05.

HkiL+LCMV and HkiL^AIDG23S + LCMV mice exhibited comparable LCMV- nAb and LASV-nAb titers. In contrast, HkiL+LASV_LII animals had higher LASV-nAb titers but lower LCMV-nAb responses than the HkiL^AIDG23S + LASV_LII group (Fig. 6C–E), resulting in a higher LASV:LCMV nAb ratio than in hypermutation-impaired HkiL^AIDG23S + LASV_LII mice (Fig. 6E).

These observations raised the question whether HkiL cells, primed by rLCMV/LASV_LII, affinity-matured to the LASV-GPC_LIV struc- ture in a directed manner, given the sequence similarity of 93.47% (Fig. 1A,B). Alternatively, LASV_LII-GPC priming may have simply diver- sified the BCR repertoire of HkiL cells. The GPC sequence of MRWV is rather distant from LASV_LIV and LCMV (71.58% and 58.11%, respectively; Fig. 1A, B) yet efficiently triggered HkiL cell proliferation (Fig. 3E). CD8 T cell-depleted TgL recipients of HkiL cells were immunized with either rLCMV/LASV_LII or rLCMV/MRWV, and control groups were left without HkiL cell transfer (Fig. 6F). When challenged with rLCMV/LASV_LIV^C four weeks later, not only HkiL+LASV_LII but also HkiL+MRWV mice con- trolled viremia at around or below detection limits, whereas controls without HkiL cell transfer were highly viremic (Fig. 6G). Consistent with comparable viral load suppression, HkiL+LASV_LII and HkiL+MRWV mice showed similar ratios of LASV: LCMV nAb at two weeks after challenge (Fig. 6H–J). Moreover, a flow cytometric analysis of HkiL cell progeny showed comparable numbers of GC B cells and MBCs in the two groups (Fig. 6K–M). Higher ASCs numbers in HkiL+MRWV mice were likely attributable to suboptimal suppression of viremia in this group (Fig. 6M, compare Fig. 6G). Taken together, these data demonstrates that immunization with a distantly related GPC (MRWV) can drive the diversification of HkiL B cells to generate clones that are effective against rLCMV/LASV_LIV^C viremia, similar to immunization with the more closely related rLCMV/LASV_LII^A.

## A clinical-stage Lassa vaccine candidate suppresses viremia independently of CD8 T cells and drives B cell affinity maturation

The Lassa fever vaccine candidate rVSVΔG-LASV-GPC, a recombinant vesicular stomatitis virus (VSV) expressing the LASV_LIV-GPC in lieu of VSVG, protects NHPs against lethal LASV challenge and currently undergoes clinical Phase II efficacy testing[18]. Here we assessed whether rVSVΔG-LASV-GPC, analogously to rLCMV/LASV_LII^A, could induce CD8 T cell-independent suppression of rLCMV/LASV_LIV^C viremia in mice. WT mice were depleted of CD8 T cells and immunized with either rVSVΔG- LASV-GPC or rVSV-EGFP as negative control (Fig. 7A). An additional control group of mice was left unimmunized. When challenged with rLCMV/LASV_LIV^C five weeks after vaccination, rVSVΔG-LASV-GPC- immunized mice developed only transient low-level viremia, which was suppressed to below detection limits within twenty days (Fig. 7B). In contrast and as expected, rVSV-GFP-vaccinated and unvaccinated controls remained comparably viremic throughout. Analogous to the findings in rLCMV/LASV_LII^A-vaccinated mice, suppression of rLCMV/LASV_LIV^C in rVSVΔG-LASV-GPC-vaccinated mice correlated with elevated LASV-GPC-binding IgG titers prior to and after challenge, whereas LASV-nAbs remained below detection limits (Fig. 7C–E; com- pare Fig. 2). Importantly, rVSVΔG-LASV-GPC-vaccinated and CD8 T cell- depleted mice suppressed not only the replication of rLCMV/LASV_LIV^C, which expresses the GPC contained in the vaccine, but they controlled viremia also when challenged with rLCMV/LASV_LI^C or rLCMV/LASV_LIII^C expressing the GPC of distinct LASV lineages (Figure S6A–D).

Next, we tested whether rVSVΔG-LASV-GPC priming rendered HkiL cells effective against rLCMV/LASV_LIV^C viremia as observed for rLCMV/LASV_LII and rLCMV/MRWV immunization. CD8 T cell-depleted TgL recipients of HkiL cells were vaccinated with either rVSVΔG-LASV- GPC or with rVSVΔG-LCMV-GPC expressing the glycoprotein of LCMV (Fig. 7F). An additional control group of BCR-restricted TgL mice was vaccinated with rVSVΔG-LASV-GPC in the absence of HkiL cell transfer. Four weeks after vaccination all groups were challenged with rLCMV/ LASV_LIV^C. Viremia in HkiL+rVSVΔG-LASV-GPC mice remained at or below detection limits whereas HkiL+rVSVΔG-LCMV-GPC animals as well as the rVSVΔG-LASV-GPC-only group were viremic throughout follow-up (Fig. 7G). On day 20 after challenge, the serum of HkiL +rVSVΔG-LASV-GPC mice exhibited LASV-nAb titers comparable to HkiL+rVSVΔG-LCMV-GPC controls (Fig. 7H), while LCMV-nAbs were significantly higher in the latter (Fig. 7I). This resulted in a significantly higher LASV_LIV:LCMV nAb ratio in HkiL+rVSVΔG-LASV-GPC mice, which correlated with viral load control as previously observed in rLCMV/LASV_LII- and rLCMV/MRWV-immunized mice (Fig. 7J, compare to Fig. 6E, J). Of note, none of the animals exhibited LASV nAb anti- bodies prior to challenge as previously observed (Figure S6E, compare to Figure S4D). Flow cytometric analysis revealed that HkiL B cells in rVSVΔG-LASV-GPC-primed mice were less abundant than in rVSVΔG- LCMV-GPC-primed controls and contained a higher proportion of MBCs, recapitulating the pattern of the antivirally effective HkiL cell response in rLCMV/LASV_LII-vaccinated mice (Fig. 7K–M; compare Fig. 4L, M). In conclusion, these findings with the clinical-stage vaccine candidate rVSVΔG-LASV-GPC recapitulated key findings and concepts established in the rLCMV/LASV_LII immunization model.

## Discussion

Recent years have seen a substantial increase in LF research and notably in clinical vaccine development. Antibody correlates are commonly used in field studies, but their value as mechanistic corre- late of protection remains uncertain. In the present study we use attenuated chimeric LCM viruses as well as rVSVΔG-LASV-GPC, an LF vaccine candidate in clinical development, to establish B cell memory and its resulting recall responses as a correlate of vaccination-induced viral load control. By providing a mechanistic basis for the use of serological assays, our findings have implications for vaccine trial readouts.

We found that the quality of the B cell recall response, in addition to the LASV-GPC-binding antibody titers at the time of challenge, predicted viral immune control. Accordingly, and in keeping with

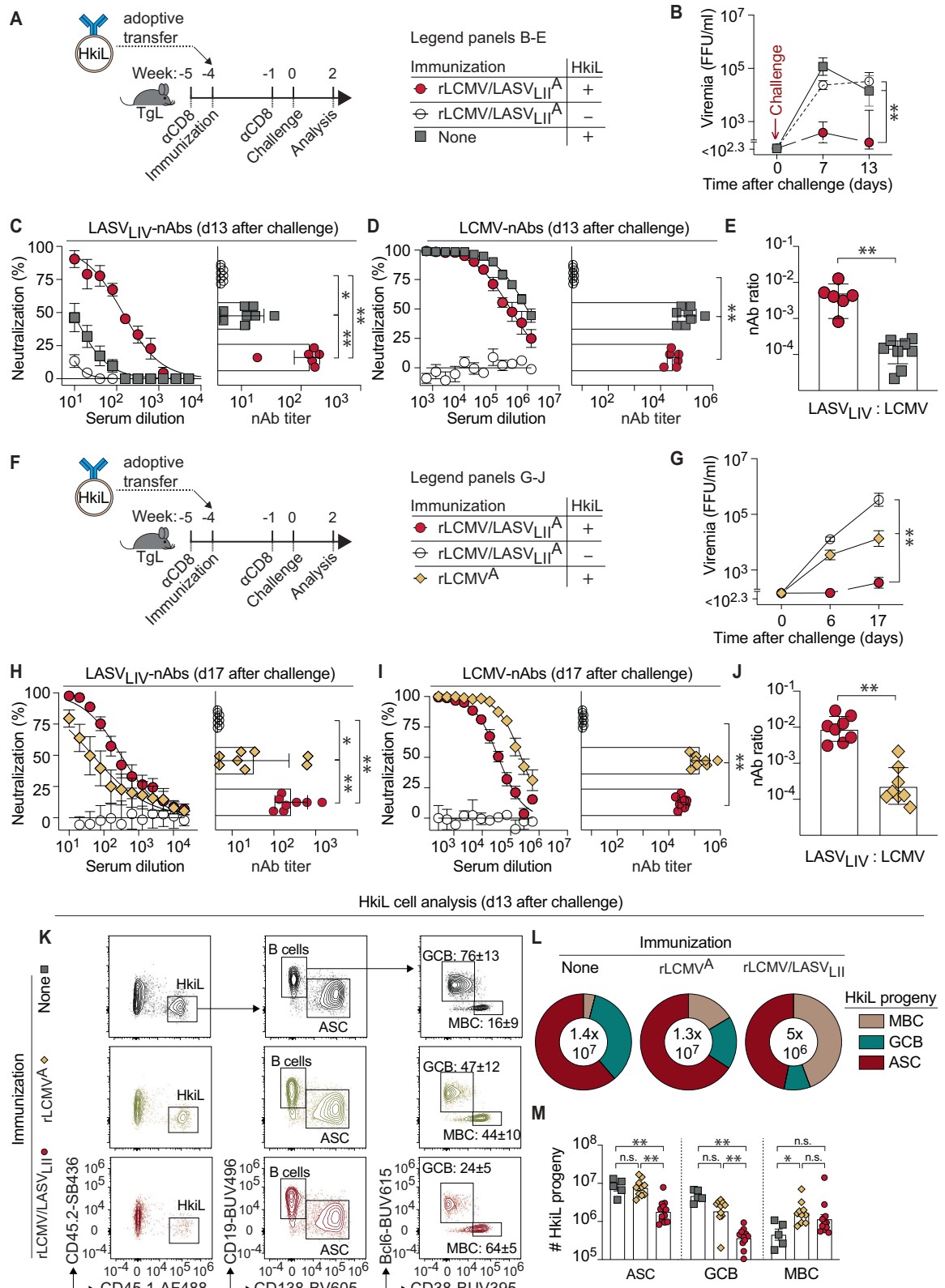

**HkiL cell analysis (d13 after challenge)**

earlier studies[19,20], a lack of LASV-nAb activity after vaccination should not be mistaken as an absence of antivirally effective humoral immunity. These concepts are supported by earlier studies on rabies- and measles virus-vectored LASV vaccine candidates in NHPs where protection from lethal LF was not only associated with T cell memory but also with accelerated LASV-neutralizing and non-neutralizing antibody responses[19,23,72,73].

While potently neutralizing mAbs were protective in passive transfer studies, active immunization commonly fails to elicit LASV-nAb activity at comparably high titers[16–24]. We found that LASV-GPC-specific secondary antibody responses, even when non-neutralizing, correlated with viral load control in CD8 T cell-depleted mice. Antibody effector functions such as ADCC (compare Figs. S2C, S4C) and antibody-dependent cell-mediated phagocytosis but also

**Fig. 4 | LASV-GPC but not LCMV-GPC priming enables control of rLCMV/LASVLIVC viremia by HkiL cells. A** Experimental outline for panels (**B–E**). At week −5, TgL mice were depleted of CD8 T cells. At wk-4 mice were immunized with rLCMV/LASVL$_{II}$A or none and given $10^4$ HkiL cells or none according to the chart. CD8 T cell depletion was repeated at wk-1. At wk0 the animals were challenged with rLCMV/LASV$_{LIV}$C. **B** rLCMV/LASV$_{LIV}$C viremia was determined over time. **C** LASV$_{LIV}$- and (**D**) LCMV-neutralizing antibody (nAb) responses at d13 with respective neutralizing titer shown on the right. **E** Ratio of LASV$_{LIV}$-nAb to LCMV-nAb titer. Panels (**B, C** right, **D** right, **E**) report combined data from two independent experiments; n = 7 (HkiL+rLCMV/LASVL$_{II}$A), n = 8 (HkiL-only) n = 7 (rLCMV/LASVL$_{II}$A-only). Panels (**C**) left and (**D**) left show data from one of the above two experiments; **n** = 3 (HkiL+rLCMV/LASVL$_{II}$A), n = 5 (HkiL-only) and n = 3 (rLCMV/LASVL$_{II}$A-only). **F** Experimental outline for panels (**G–J**). This experiment was conducted analogously to (**A–E**), with groups that were immunized and received cell transfer as indicated in the chart: Panels (**G, H**) right, I right and J report combined data from two independent experiments; n = 8 (HkiL+rLCMV/LASVL$_{II}$A), n = 8 (HkiL+rLCMV), n = 6 (rLCMV/LASVL$_{II}$A-only). Panels (**H**) left and I left show data from one of the two above experiments; n = 5 (HkiL+rLCMV/LASVL$_{II}$A), n = 5 (HkiL+rLCMV), n = 3 (rLCMV/LASVL$_{II}$A-only). **G** rLCMV/LASV$_{LIV}$C viremia over time. **H, I** LASV$_{LIV}$- and LCMV-nAb response with respective neutralizing titer shown on the right. **J** Ratio of LASV$_{LIV}$-nAb to LCMV-nAb titer. (**K-M**) Analysis of HkiL cell (CD45.1$^+$) progeny at wk2. (**K** Exemplary FACS plots from n = 5 (none), n = 3 (rLCMV$^A$), and n = 3 (rLCMV/LASV$_{II}$A) mice, showing the gating strategy for the populations enumerated in (**L, M**). Numbers in plots indicate the percentage of gated cells as mean ± SD. GC B cells (GCBs) were identified as CD19$^+$ GL7$^+$ CD38$^{lo}$, antibody-secreting cells (ASCs) as CD19$^{lo}$CD138$^+$, memory B cells (MBC) as CD19$^+$GL7$^-$CD38$^{hi}$. **L** Relative representation of HkiL progeny. **M** Absolute numbers of HkiL progeny in the spleen. Panels L and M report combined data from two experiments; n = 5 (none), n = 10 (rLCMV$^A$), and n = 12 (rLCMV/LASV$_{II}$A). Bars denote the mean ± SD with symbols representing individual mice, symbols on antibody titer curves represent the mean ± SEM. Two-way ANOVA with Sidak's post-test was used in (**B, G**) to compare rLCMV/LASV$_{LII}$+HkiL mice against all other groups, one-way ANOVA with Tukey's post-test was conducted in (**C, D, H, I, M**), unpaired two-tailed Student's t-test was used in (**E, J**). **\*\****p* ≤ 0.01; **\****p* ≤ 0.05; ns: *p* > 0.05.

complement-mediated virion lysis may have contributed to this effect[19,20,74]. Still, LASV-GPC binding alone was not sufficient to predict antiviral efficacy. This is exemplified by the KL25 antibody, which fails to suppress rLCMV/LASV$_{LIV}$C viral loads albeit directed against a neutralizing epitope on the viral GP-1. These observations suggest that quality criteria exists that define the ability of serum antibody responses to contain viremia. Such criteria likely include affinity/avidity of the individual antibody[75] as well as the compound effect of simultaneous binding to different epitopes and antigenic regions on GPC, a phenomenon commonly observed in other viral diseases[37,76].

Affinity maturation is generally required to generate nAbs to OW arenaviruses and is widely accepted to improve antibody quality even if viral neutralizing thresholds are not reached[66,77]. The use of affinity maturation-impaired HkiL$^{AIDG23S}$ B cells supports the notion that affinity maturation is important to generate an antivirally effective LASV-GPC-specific B cell repertoire. In a complementary approach, priming with the distantly related GPC of MRWV revealed that diversification of the repertoire, in addition to mere affinity maturation, is contributing to the antiviral efficacy of B cell memory. While demonstrated here for monoclonal HkiL B cells, the same principles likely apply to polyclonal responses. Besides improved affinity, the augmented clonal diversity of a secondary B cell response to LASV-GPC is expected to offer broader coverage of LASV lineages and variants.

The role of OW arenavirus cross-reactive immunity has been extensively studied at the level of T cells[78]. Our study establishes the importance of cross-reactive B cell responses against LASV-GPC and is thus of practical relevance given its role as primary vaccine antigen. There has been concern that the GPC heterogeneity between different LASV lineages may preclude the success of a universal LASV vaccine candidate. The present findings suggest that B cell memory to LASV-GPC is broader than commonly thought, arguing that B cell-based suppression of LASV loads can be lineage-agnostic. Beyond LF vaccines, it is well established that non-pathogenic OW arenaviruses can induce cross-protective immunity against LF. The present findings suggest that besides cell-mediated immunity also GPC-specific B cell memory may have contributed to these observations. LCMV seroprevalence can be high in sub-Saharan Africa[79,80], and close relatives of LASV such as MOBV co-circulate in geographic areas where LASV is endemic. It remains unknown, however, to which extent human exposure to these viruses may contribute to a mostly mild course of LASV infection in the field. This work establishes B cell memory against OW arenaviruses as a mechanistic correlate of viral load control. Vaccines inducing potent B cell memory, ideally in conjunction with CD8 T cell immunity, may therefore offer broad immunity against a wide range of LASV lineages.

Our study has limitations including the use of recombinant LCMV/LASV, which is only an imperfect surrogate of LASV and does not induce hemorrhagic fever in mice. The latter context can, for example, result in immunosuppressive effects, which are insufficiently recapitulated in the models used here and may influence secondary immune responses. It will therefore be important that the key concepts derived from this work be independently confirmed and corroborated in commonly used LASV challenge models. More generally, the humoral immune responses of humans and NHPs to LASV-GPC can differ from responses in mice, which is due at least in part to the numerically smaller lymphocyte repertoire of the latter. Of further note, CD8 T cell depletion prevents both the clinical and biochemical manifestations of disease in mouse models of LF as well as in LCMV-infected mice[81–86]. While essential to distinguish between CD8 T cell- and antibody-mediated viral load control, CD8 T cell depletion therefore eliminated disease-related parameters as usable readouts in our study. Similarly, Fcγ receptor-mediated effector functions are essential for antibody-mediated CD8 T cell depletion[87], precluding the use of Fcγ receptor-deficient mice to assess the Fcγ receptor-dependence of antibody-mediated viral load control in our experimental setting. We further acknowledge that, while establishing the ability of LASV-GPC-specific B cell memory to suppress viremia, the design of our study does not differentiate between recall responses by bona fide memory B cells and the refueling of long-lived germinal center reactions still present at the time of challenge. Both subsets of B cells are likely to persist for prolonged periods of time after vaccination with live-attenuated viruses as used in our study, and both subsets may contribute to suppression of viral loads. A further limitation of this work consists in the reliance on monoclonal HkiL B cells, which were useful for dissecting mechanistic aspects of the clonal evolution of LASV-GPC−specific antibody responses but may not fully represent the physiological human B-cell repertoire. Importantly also, LASV-nAbs in the serum of HkiL B cell recipients should merely be interpreted as a surrogate of affinity-matured and LASV-GPC-adapted HkiL cell responses. We acknowledge that the LASV-neutralizing subset of HkiL cell-derived antibodies likely corresponds to only a small fraction of highly matured antibodies. Suppression of rLCMV/LASV$_{LIV}$C viremia in vaccinated HkiL cell recipients was presumably mediated by a larger pool of non-neutralizing but LASV-GPC-adapted KL25 variants.

## Methods

### Animals and ethics statement

Mouse experiments were carried out at the University of Basel in accordance with the Swiss law for animal protection and with authorization from the Cantonal veterinary office Basel-Stadt. C57BL/6 J (WT) mice were originally purchased from Charles River and were

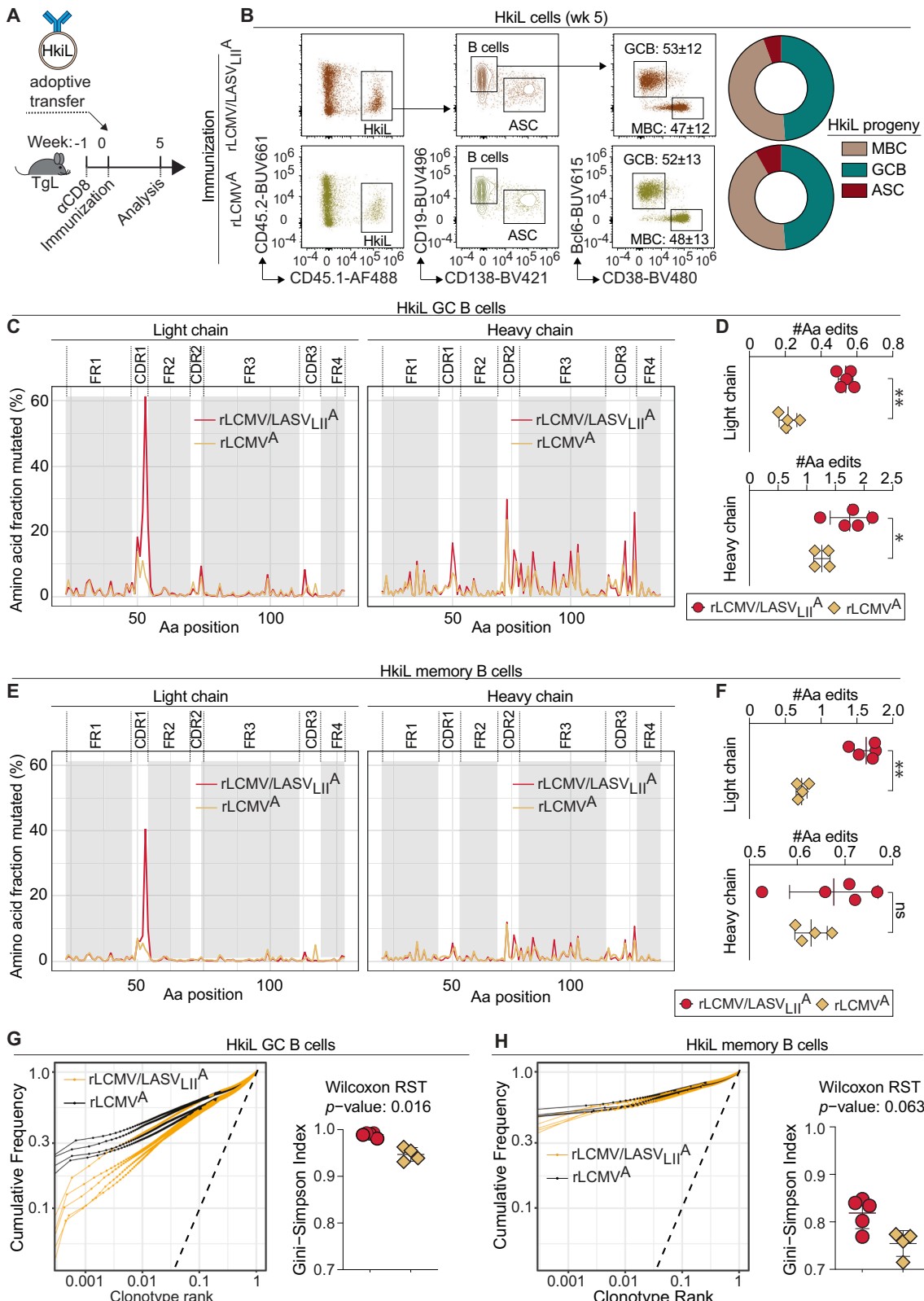

bred at the ETH Phenomics Center (EPIC) Zurich. As donors of monoclonal LCMV-specific BCR-expressing B cells (HkiLs) HkiL-RAG[−/−] mice[66] hemizygous for the knocked-in immunoglobulin heavy and light chain loci were used. For simplicity these animals are referred to as HkiL mice in the manuscript. TgL mice were used as recipients for HkiL B cells[77,88]. The AID[G23S] strain[77] was intercrossed with HkiL-RAG[−/−] mice to obtain HkiL-RAG[−/−]-AID[G23S] homozygous for the AID[G23S] allele

and hemizygous for the knocked-in immunoglobulin heavy and light chain loci. For simplicity these animals are referred to as HkiL-AID[G23S] in the manuscript. sIgM[−/−]AID[−/−] mice have been reported[89] and were obtained by cross-breeding AID[−/−][90] mice with sIgM[−/−] mice[91]. All genetically modified mice were on a C57BL/6 J background. Mice were housed under specific-pathogen-free (SPF) conditions during breeding and experiments with a 12 h – 12 h light – dark cycle at a temperature of

**Fig. 5 | The immunizing GPC determines the extent of HkiL cell hypermutation and clonal diversification. A** Experimental outline for the analyses in panels (**B**–**H**). At week −1, TgL mice were depleted of CD8 T cells. At wk0 they were given $10^4$ HkiL cells and were immunized with either rLCMV/LASVL$_{II}$$^A$ ($n = 5$) or rLCMV$^A$ ($n = 4$). At wk5 HkiL cell progeny in spleen were analyzed. **B** HkiL cell differentiation as determined by flow cytometry. Concatenate FACS plots ($n = 4$-5 per group) show the gating strategy for the relative repartition of HkiL progeny into GC B cell (GCB), memory B cell (MBC) and antibody-secreting cell (ASC) subpopulations as displayed in the pie chart and pre-gated as shown in Figure S3A. Numbers in plots indicate the percentage±SD of gated cells. (**C**, **E**) Occurrence of hypermutations along the V(D)J sequence of HkiL GCBs (**C**) and MBCs (**E**) as determined by single cell RNA sequencing (scRNAseq). **D**, **F** Average number of amino acid exchanges in the complementarity-determining regions (CDRs) of HkiL GC B cell (**D**) and MBC (**F**) light and heavy chains as determined by scRNAseq. **\*\***$p \leq 0.01$; **\***$p \leq 0.05$; n.s.:

$p > 0.05$ by unpaired two-tailed Student's t-test. **G**, **H** Curves show the proportion of cells per clonotype for each animal, arranged from left to right by decreasing clonotype abundance. The dashed bisecting line corresponds to a hypothetical uniform clonotype abundance. Deviation from the dashed line indicates inhomogeneity of clonotype abundance. Average Gini-Simpson diversity index of individual animals and resulting two-tailed Wilcoxon Rank Sum Test- (RST-) p-values are shown. HkiL GC B cells are shown in (**G**), HkiL MBCs in (**H**). To level out differences in the resolution of clonal frequencies, we randomly down-sampled cells per animal and cell type to the lowest observed total number of cells. Shown is the average Gini-Simpson diversity index per animal and cell type over 100 random draws. Curves in (**C**, **E**) show the cumulated data from all animals in the respective group, symbols in (**D**, **F**) and lines in (**G**, **H**) represent individual animals. Symbols in (**D**, **F**, **G** right and **H**) right shown individual mice with the mean ± SD indicated.

$22 \pm 2\,°C$ and $55 \pm 10\%$ humidity and in keeping with all federal regulations. Mice in experimental groups were sex- and age-matched. All mice were typically in the age range of 8–14 weeks at the start of the experiment and groups were sex- and age-matched. In keeping with our ethics protocol, sex has not been considered as an independent parameter in our study design. Animals of both genders were used to reduce the number of animals bred for research purposes. The groups were not randomized and the experiments were not conducted in a blinded fashion.

### Virus infections, CD8 T cell depletion, adoptive B cell transfer and blood sampling

For infections and immunizations with LCMV and derived viruses a dose of $\geq 10^7$ focus-forming units (FFU) was administered intravenously (i.v.) in a volume of 200 µl, except for LCMV-Armstrong ($10^3$ FFU i.v.). The VSV-based viruses were administrated i.v. at a dose of $10^6$ PFU (plaque-forming units). For CD8 T cell depletion, a total of 0.4 mg of the YTS169.4 (mouse IgG2a) was administrated as initial dose whereas a dose of 0.2 mg was used immediately prior to or after challenge, as indicated. In the experiment to Fig. 5 and S5 the YTS169.4 depletion antibody was delivered by an AAV vector administered intramuscularly to mice at a dose of $5 \times 10^{10}$ vp[89]. CD8 T cell depletion was confirmed by flow cytometric analysis from blood. For adoptive cell transfer of B cells from HkiL and HkiL-AID$^{G23S}$ mice, spleens from the respective donor animals were collected in HBSS, macerated to a single cell suspension and strained through a 70 µm mesh. Recipient mice received $10^4$ HkiL B cells corresponding to ~500 engrafted cells per spleen assuming an engraftment rate of 5%[92]. In the experiments to Fig. 3D and S3D HkiL cells were labelled with carboxyfluorescein succinimidyl ester (CFSE; CellTrace, Invitrogen) and $2 \times 10^5$ cells were adoptively transferred to allow visualization of adoptively transferred HkiL B cells without prior expansion. Blood was collected at the indicated timepoints from the tail vein. To determine viremia, one drop of blood was mixed with 950 µl of balanced salt solution (BSS) supplemented with 1 IE/ ml Heparin-Na (B. Braun, B01AB01) and stored at −80 °C until analysis. For serological analysis, blood was collected into Multivette 600 Serum gel tubes (Starstedt), processed for serum separation as indicated by the vendor and stored at −20 °C for subsequent use in virus neutralization assay and ELISA.

### Cell lines

FreeStyle 293-F suspension cells were obtained from Gibco/ Thermo-Fisher (R790-07) and cultured in HyClone CDM4HEK293 media (Cytiva, SH30858.02; supplemented with 4 mM GlutaMax (Gibco, 35050038)). Vero E6 cells were purchased from ECACC (85020206) and cultured in DMEM supplemented with 10% FBS (Fetal Bovine Serum). BHK-21 cells (clone 13; 85011433) were purchased from the European Collection of Authenticated Cell Cultures (ECACC) and were cultured in DMEM (Sigma-Aldrich) supplemented with 10% FBS, 1 mM HEPES, 100 mM sodium pyruvate, and 1x Tryptose Phosphate Broth

(Gibco). BHK-21 cells expressing the LCMV GPC have been described in ref. 93 and were cultured analogously to BHK-21 cells but with the addition of 2 µg/ml puromycin to the medium. NIH/3T3 cells (CRL-1658), obtained from the American Type Culture Collection (ATCC), were cultured in DMEM containing 10% FBS. All cells were cultured at 37 °C in an atmosphere of 5% $CO_2$. All cell lines were tested for mycoplasma at regular intervals and were confirmed negative.

### Glycoprotein complex (GPC) sequences and genealogy analyses

The GPC sequences corresponding to the following virus strains (GenBank accession numbers) were used: Lassa virus Pinneo (AY628207.1), Lassa virus LII 803213 (AF181854.1), Lassa virus LIII GA391 (OL774861.1), Lassa virus LIV Josiah (NC_004296), LCMV WE (FJ607036), Lassa virus LIII GA391 (OL774861.1), Lujo virus (NC_012776.1), Mopeia virus (AY772170.1), Mobala virus (NC_007903.1), and Merino Walk virus (NC_023764.1). To determine the genetic relatedness of the above arenavirus GPCs and their sequence homology at the amino acid level we used MegalignPro (DNAStar) with Clustal W for sequence comparisons using the BIONJ algorithm[94]. cDNAs of the GPC sequences were synthesized by Genscript.

### Generation, propagation and titration of chimeric LCMV- and VSV-based viruses

Genetically engineered rLCMV were generally based on Armstrong Clone 13 as a backbone (Genbank accession number DQ361065) unless specified otherwise. As sole exception, LCMV-ARM used in this study was a recombinant LCMV-Armstrong-based virus[95] (expressing the glycoprotein of the WE strain (Genbank accession number AJ297484). The chimeric LASV/LCMV GPCs and LCMV/LASV GPCs were generated by PCR cloning using the in-Fusion® Snap Assembly Master Mix kit (638946 Takara Bio Inc). Plasmids were subject to Sanger sequencing to confirm sequence integrity, and they were integrated into S segment expression plasmids[96]. Infectious LCMV-based viruses (rLCMV) incorporating various glycoprotein sequences instead of the natural LCMV Cl13 GPC were generated from cDNA. Working stocks were obtained by passaging of the viruses on BHK-21 cells at a multiplicity of infection (MOI) of 0.01. All rLCMV viruses were subsequently passaged on FreeStyle 293-F suspension cells at a multiplicity of infection (MOI) of 0.001. Infectious titers of LCMV stocks and mouse blood samples were determined by focus forming assay on 3T3 cells.Three-fold serial dilutions of virus-containing samples were incubated with NIH 3T3 cells (ATCC) in 96-well plates, followed by 3 hours of incubation at 37 °C. Then, overlay (1% methylcellulose in DMEM) was added and the cultures were incubated for two days. On the third day, supernatant was removed, the cells were fixed with 4% paraformaldehyde and permeabilized (1% Triton X-100 in PBS). After blocking (5% FCS), infectious foci were visualized using VL4 rat-anti-LCMV-NP antibody (1:50) and secondary HRP-conjugated goat-anti-rat-IgG (Jackson Immunoresearch) (1:500), followed by an enzymatic color reaction.

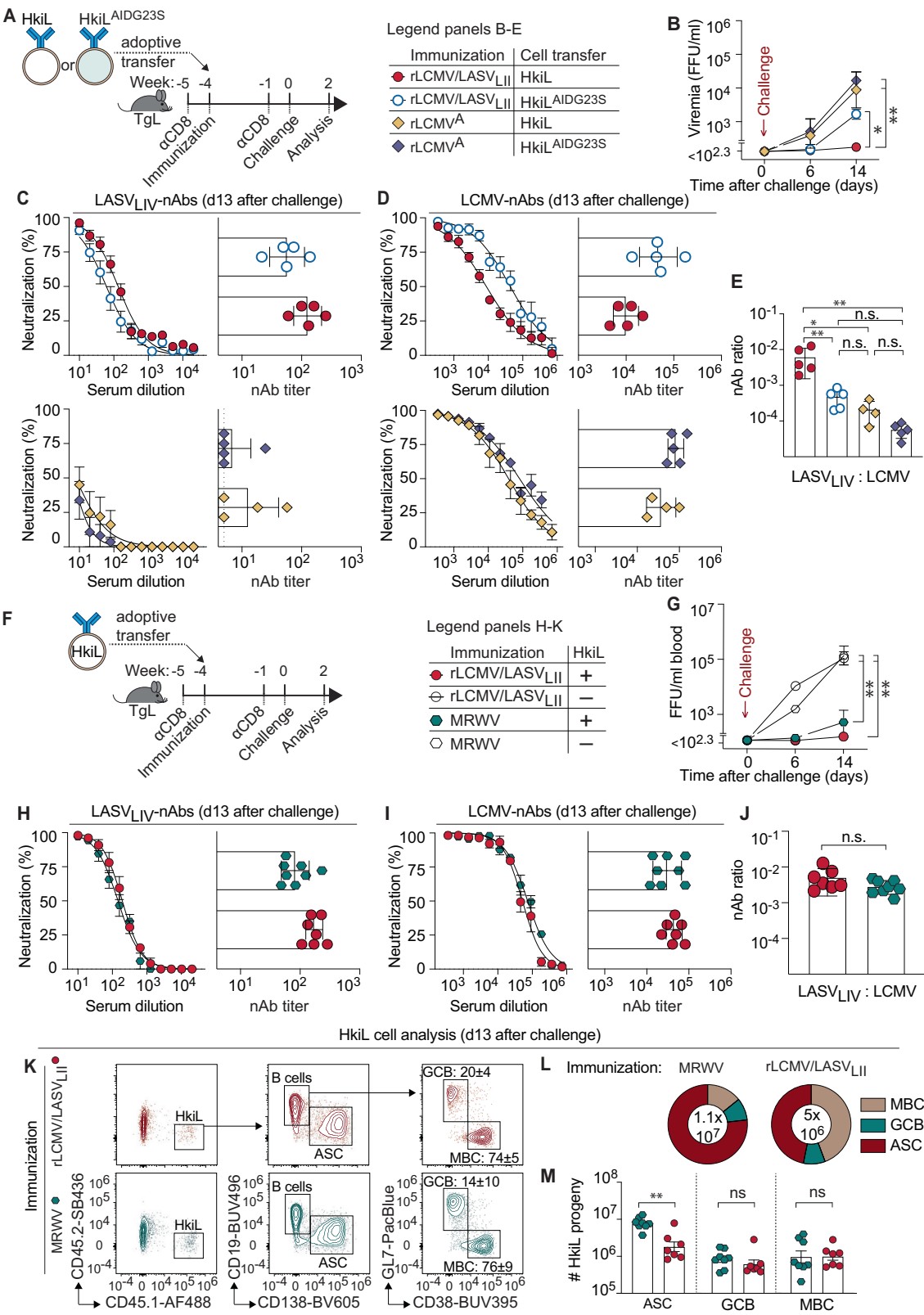

Recombinant vesicular stomatitis viruses (rVSV) were engineered to express instead of their natural VSV-G surface glycoprotein either the LASV$_{LIV}$-GPC (NC_004296) or the LCMV strain WE GPC (FJ607036). A GFP-transgenic VSV expressing GFP from an additional transcription start–stop cassette has been described in ref. [97], and a VSV expressing the Zaire Ebolavirus glycoprotein instead of VSV-G (VSVΔG/ZEBOVGP[98] was generously provided by Viktor Volchkov, INSERM, France). The infectious viruses were rescued from cDNA following established procedures[99]. Viral titers were determined by plaque forming assay on Vero E6 cells. Glycoprotein-deficient rVSVΔG-EGFP reporter virus for virus neutralization assays was grown in HEK 293 T cells transiently expressing the respective glycoprotein and titrated on Vero E6 cells based on eGFP expression at the single cell level using a fluorescence microscopy scanner[99].

**Fig. 6 | Suppression of rLCMV/LASVLIVC viremia requires HkiL cell receptor diversification by hypermutation. A** Experimental outline for panels (B-E). At wk-5, TgL mice were depleted of CD8 + T cells. At wk-4 the animals were immunized and received HkiL or HkiL$^{AIDG23S}$ cell transfer according to the chart. CD8 T cell depletion was repeated at wk-1. At week 0, animals were challenged with rLCMV/LASV$_{LIV}$$^C$. **B** rLCMV/LASV$_{LIV}$$^C$ viremia at the indicated timepoints. **C** LASV$_{LIV}$- and (**D**) LCMV-nAb responses at wk2 with respective neutralizing titer on the right. **E** Ratio of LASV$_{LIV}$-nAb to LCMV-nAb titer. Panels B-E report data from $n = 4$ (HkiL+rLCMV$^A$) and $n = 5$ (other groups) mice per group. **F** Experimental outline for panels (**G–M**). At wk-5, TgL mice were depleted of CD8 T cells. At wk-4, mice were immunized and received 10$^4$ HkiL cells as indicated in the chart. **G** rLCMV/LASV$_{LIV}$$^C$ viremia at the indicated timepoints. **H, I** LASV$_{LIV}$- and LCMV-nAb responses at wk2 with respective neutralizing titer on the right (J) Ratio of LASV$_{LIV}$-nAb to LCMV-nAb titer. (K-M) Analysis of HkiL cell (CD45.1$^+$) progeny at wk2. (**K**) Exemplary FACS plots from $n = 3$ (HkiL+rLCMV/LASV$_{LII}$$^A$) and $n = 5$ (rLCMV/MRWV) mice showing the gating strategy for the populations enumerated in (L) and (M). Numbers in plots indicate the

percentage of gated cells as mean ± SD. GC B cells were identified as CD19$^+$GL7$^+$CD38$^{lo}$, antibody-secreting cells (ASCs) as CD19$^{lo}$CD138$^+$, memory B cells (MBC) as CD19$^+$GL7$^-$CD38$^{hi}$. **L** Relative representation of HkiL progeny. **M** Absolute numbers of HkiL progeny in the spleen and symbols follow the denotation given in (**F**). Panels (**L, M**) report on the numerical proportion of HkiL cell subsets (**M**) and their absolute counts (**L**). In (**M**) the total number of cells analyzed is indicated in the center of the pie chart. Panels (**G–I, L, M**) show $n = 7$ (HkiL+rLCMV/LASV$_{LII}$$^A$), $n = 8$ (HkiL+rLCMV/MRWV) and $n = 6$ (rLCMV/LASV$_{LII}$$^A$-only) from two combined experiments and $n = 3$ (rLCMV/MRWV only) from one experiment. Bars denote the mean ± SD with symbols representing individual mice, symbols on curves represent the mean ± SEM. One out of two similar experiments is shown for (**B–E**) while in (**G–J**) data were pooled from two independent experiments. Two-way ANOVA with Šídák's post-test was performed in (**B, G**) to compare the rLCMV/LASV$_{LII}$+HkiL group against all other groups, one-way ANOVA with Tukey's post-hoc test was conducted in (**M, E**), two-tailed Welch's $t$-test was used in (**C, D, H, I, J**). **$p ≤ 0.01$; *$p ≤ 0.05$; n.s. $p > 0.05$.

## Viral sequence determination

For viral RNA isolation and sequencing, BHK21 cells in a 6-well plate with 2 ml of BHK-21 medium (see above) were infected with 50 µl of blood samples pre-diluted in 1:20 in balanced salt solution containing 1 IU/ml heparin. Supernatant was collected two days later. Viral RNA was extracted from virions in the supernatant using the QIAamp Viral RNA Mini Kit (QIAGEN) in accordance with the manufacturer's instructions. cDNA synthesis of the LASV GPC was performed with the SuperScript IV first strand synthesis system (Thermofisher), followed by PCR using Phusion polymerase (New England Biolabs) and primers 5'-GATCCTAGGCTTTTTGGATTGCG-3' and 5'-AAGAAAGA-GATCACCCCGCACTGT-3'. The PCR products were run on 1.5% agarose gel, amplicons of the correct size were excised, purified using the QIAquick Gel Extraction Kit (QIAGEN), and subject to Sanger sequencing (Microsynth AG, Switzerland).

## Flow cytometry

Spleens were collected into RPMI, which was supplemented with 10% FBS, 5 mM HEPES, 50 µM 2-mercaptoethanol and adjusted to mouse osmolarity[100]. After collection spleens were macerated to a single cell suspension and strained through a 70 µm mesh. Surface marker staining was performed in FACS buffer (PBS + 2%FBS, 1 mM EDTA; adjusted to mouse osmolarity) at 4 °C for 20 min. For live/dead staining of fixed samples Zombie UV fixable viability kit was used (Biolegend) or DAPI when measuring unfixed samples. Cells were fixed using 2% paraformaldehyde (PFA) in PBS for 10 min at room temperature. Alternatively, for transcription factor detection the Foxp3/Transcription Factor Staining Buffer Set (eBioscience™) was used. Transcription factors were stained at 4 °C for 12–18 h. Fluorescently labelled cells were measured on an LSRFortessa (Becton Dickinson) or on an Aurora (Cytek) flow cytometer. When staining cells in peripheral blood, samples were fixed and lysed by adding 1 ml/sample of eBioscience 1-step Fix/Lyse Solution and incubating at RT for 5 min. The reaction was stopped by adding FACS-buffer. A list of antibodies detailing their target molecules, fluorophore conjugate, dilution for use, clone designation, provider and catalogue number is provided as Supplementary Table I. Viral epitope-specific CD8 T cells were identified using MHC class I tetramers loaded with the LCMV nucleoprotein-derived immunodominant epitopes NP396-404 (FQPQNGQFI) and NP205-212 (YTVKYPNL), which were used at a 1:50 dilution for staining and were supplied by the NIH Tetramer Core Facility and the University of Lausanne Tetramer Core Facility, respectively. For the plots depicted in Figure S2D the cells were pre-gated on B220$^-$CD8$^+$ lymphocytes.

## Single-cell RNA sequencing

Splenic single cell suspensions were generated by smashing spleens between frosted edges of microscopy slides (Carl Roth, catalog no.

1879.1) into Opti-MEM (Gibco, catalog No 11058021) supplemented with 1% penicillin/streptomycin (Gibco, catalog no. 10378016), 1% MEM Non-Essential Amino Acids Solution (Gibco, catalog no. 11140050), 1% MEM amino acids solution (Gibco, catalog no. 11130051), 2 mM L-glutamine (Gibco, catalog no. 25030081), 1% HT supplement (Gibco. catalog no. 11067030), 50 µM 2-Mercaptoethanol (Gibco, catalog no. 31350010), 10% Fetal Bovine Serum (Gibco, catalog no. A5256701), 1 mM sodium pyruvate (Gibco, catalog no. 11360070), 10 mM HEPES (Gibco, catalog no. 15630080) and 1% insulin-transferrin-selenium-ethanolamine (catalog no. 51500056, Gibco), adjusted to mouse osmolarity. After filtering through a 30 µm MACS SmartStrainers (Milteny, catalog no. 130-098-458) single cell suspensions were stained with biotinylated antibodies against Ter119 (clone TER-119, catalog no. 116204, BioLegend), CD3 (clone 17A2, catalog no. 100244, BioLegend), CD4 (clone RM4-5, catalog no. 100508, BioLegend), CD45.2 (clone 104, catalog no. 109804, BioLegend), NK1.1 (clone PK136, catalog no. 108704, BioLegend) and Thy1.1 (clone OX-7, catalog no. 202510, BioLegend) for 20 mins in FACS Buffer (PBS supplemented with 1% penicillin-streptomycin (Gibco, catalog no. 10378016), 1% MEM Non-Essential Amino Acids Solution (Gibco, catalog no. 11140050), 2 mM L-glutamine (Gibco, catalog no. 25030081), 20 mM HEPES (Gibco, catalog no. 15630080), 1% MEM amino acids solution (Gibco, catalog no. 11130051), 1% HT supplement (Gibco, catalog no. 11067030), 50 µM 2-mercaptoethanol (Gibco, catalog no. 31350010) 10% Fetal Bovine Serum (Gibco, catalog no. A5256701), 1 mM sodium pyruvate (Gibco, catalog no. 11360070), 1% insulin-transferrin-selenium-ethanolamine (Gibco, catalog no. 51500056) and 2 g/L Glucose Solution (ThermoFisher, catalog no. A2494001) adjusted to mouse osmolarity). Negative enrichment of CD45.1 HkiL B cells was performed using the EasySep™ Mouse Streptavidin RapidSpheres™ Isolation Kit (catalog no. 19860, STEMCELL Technologies) according to manufacturer's instructions. Cells were washed once and individual samples were tagged by staining with anti-mouse MHC-H2 class I antibody (clone M1/42 Becton Dickinson, catalog no. 626545) in FACS Buffer according to the manufacturer's instructions and stained with antibodies against CD138 (Milteny, catalog no. 130-108-989), CD45.1 (clone A20, catalog no. 110718, BioLegend), B220 (BioLegend, clone RA3-6B2, catalog no. 103246), TCR-β (clone H57-597, catalog no. 109224, BioLegend) and CD45.2 (clone 104, catalog no. 757503, Becton Dickinson). Live-cell staining was done by incubation with DAPI (BioLegend, catalog no. 422801) prior to acquisition. CD138$^-$B220$^+$CD45.1$^+$CD45.2$^-$DAPI$^-$ HkiL B cells were purified by sorting on a FACSAria III cell sorter (Becton Dickinson) or on a BD FACSDiscover™ S8 Cell Sorter (Becton Dickinson).

After sorting, single cell suspensions were pooled equally. Cell number and viability were determined using a Cellometer K2 Image Cytometer (Nexcelom Bioscience, Cellometer K2) by using the

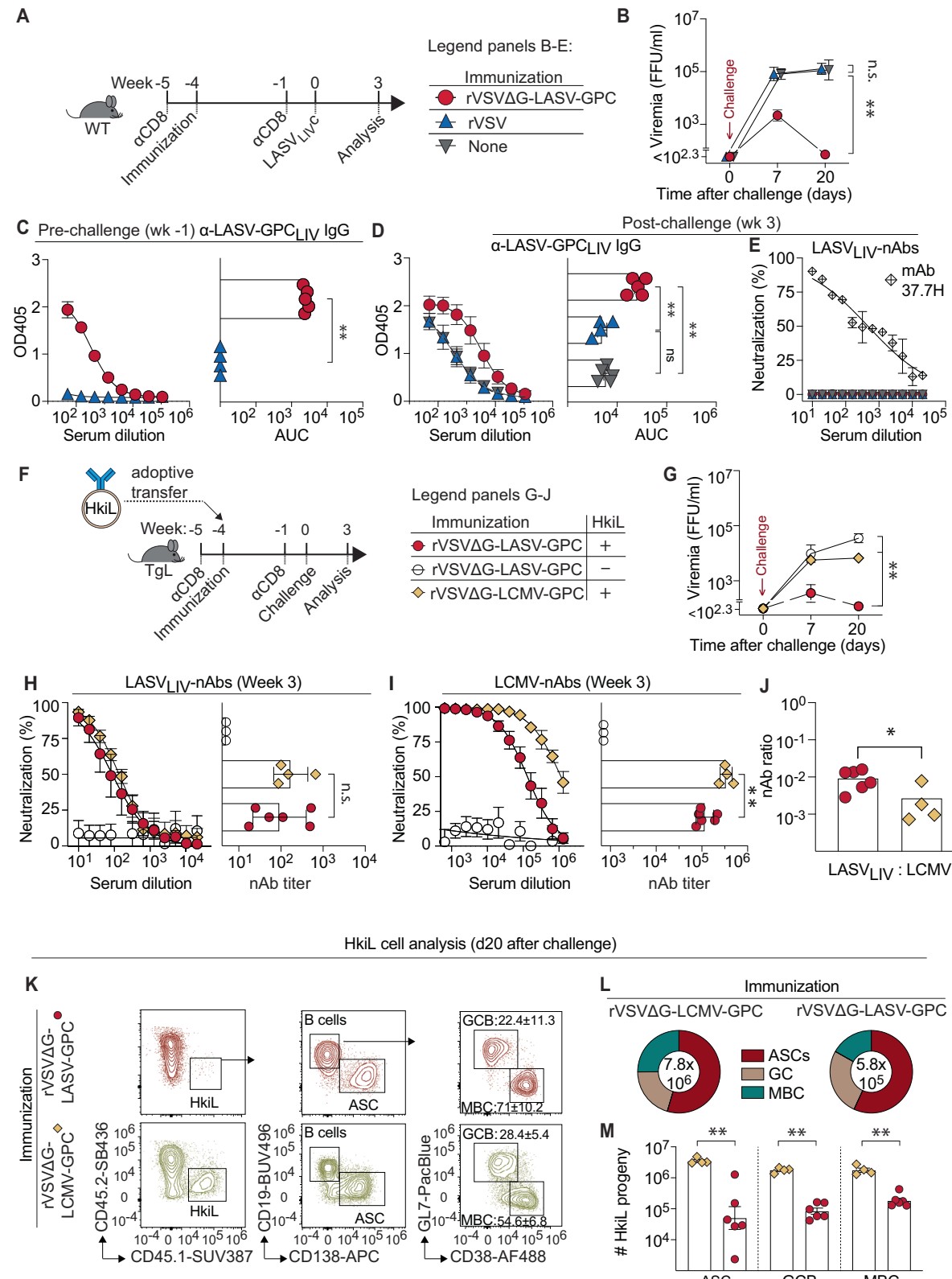

ViaStain AOPI Staining Solution (Cat#CS2-0106-5ml, Nexcelom Bioscience) and PD100 cell counting slides (Cat# CHT4-PD100-003, Nexcelom Bioscience). The pooled cell suspension was loaded into two lanes of a BD Rhapsody cartridge, targeting a capture of approximately 100,000 cells per lane. Libraries were prepared following the BD Rhapsody System Mouse TCR/BCR, Targeted mRNA and Sample Tag Library Preparation Protocol (Protocol # 23.24513(01)).

Library quality control (QC) was performed using a 5200 Fragment Analyzer 12-capillary system (Agilent, cat. No M5310AA) with HS NGS Fragment Kit (1-6000 bp) reagents (Agilent, cat. No: DNF-474-1000). Library quantification was carried out using a Qubit 4 Fluorometer (Thermo Fisher Scientific, cat. No: Q33238) and the Qubit 1X dsDNA HS Assay Kit (Thermo Fisher Scientific, Cat#: Q33231).

**Fig. 7 | A clinical-stage Lassa vaccine candidate suppresses viremia independently of CD8 T cells and drives B cell affinity maturation. A** Outline of the experiment in panels (**B**–**E**). At wk-5, WT mice were depleted of CD8 T cells. At wk-4, mice were immunized with either rVSVΔG-LASV-GPC ($n = 5$), control rVSV vector (rVSV-EGFP control; $n = 4$)) or they were left unvaccinated ($n = 5$) as indicated in the chart. CD8 T cell depletion was repeated at wk-1 and at week 0, the animals were challenged with rLCMV/LASV$_{LIV}$$^C$. **B** rLCMV/LASV$_{LIV}$$^C$ viremia over time. **C, D** anti-LASV$_{LIV}$-GPC IgG titers at wk-1 and at wk3. **E** LASV$_{LIV}$-nAb at wk3 with mAb 37.7H included as positive control. **F** Outline of the experiment in panels (**G**–**M**). TgL mice were depleted of CD8 T cells at wk-5. At wk-4 they were immunized and given HkiL cells as indicated in the chart, were again depleted of CD8 T cells at wk-1 and challenged with rLCMV/LASV$_{LIV}$$^C$ at wk 0. **G** rLCMV/LASV$_{LIV}$$^C$ viremia over time. **H, I** LASV$_{LIV}$- and LCMV-nAb responses at wk3 with respective neutralizing titer on the right. **J** Ratio of LASV$_{LIV}$-nAb to LCMV-nAb titer. **K**–**M** Analysis of HkiL cell (CD45.1$^+$) progeny at wk3. **K** Exemplary FACS plots depicting the gating strategy for the populations enumerated in (**L**, **M**). Numbers in plots indicate the percentage of gated cells as mean ± SD. Germinal center B cells (GC) were identified as CD19$^+$GL7$^+$CD38$^{lo}$, antibody-secreting cells (ASCs) as CD19$^{lo}$CD138$^+$, memory B cells (MBC) as CD19$^+$GL7$^-$CD38$^{hi}$. **L** Relative representation of HkiL progeny. **M** Absolute numbers of HkiL progeny in the spleen. Symbols follow the denotation in (**F**). $n = 6$ (HkiL+rVSVΔG-LASV-GPC), $n = 4$ (HkiL+rVSVΔG-LCMV-GPC), $n = 3$ (rVSVΔG-LASV-GPC only). Bars denote the mean ± SD with symbols representing individual mice, symbols on curves represent the mean ± SEM. **K** shows representative FACS plots from the indicated group, the percentage of gated cells is indicated as mean ± SD. (**L, M**) show the mean of the numerical proportion of HkiL cell subsets (**L**) and their absolute counts (**M**). In (**L**) the total number of cells analyzed is indicated in the center of the pie chart. Panels (**B**–**M**) show one out of two similar experiments. Two-way ANOVA with Šídák's post-test was performed in (**B, G**) to compare the rVSVΔG-LASV-GPC and the HkiL+rVSVΔG-LASV-GPC groups, respectively, one-way ANOVA with Tukey's post-hoc test was conducted in (**D, M**), two-tailed Welch's *t*-test in (**C, H, I, J**). **$p \leq 0.01$; *$p \leq 0.05$; n.s. $p > 0.05$.

The prepared libraries were sequenced on an Aviti System (Element Biosciences). Each library pool was sequenced across two lanes of a High Output 2 × 75 cycles Flowcell (Element Biosciences, Aviti 2 × 75 Sequencing Kit Cloudbreak FS High Output, Cat#: 860-00015), totaling four lanes (two full sequencing runs) per library pool.

### Analysis of single cell RNA sequencing data

The sequencing data were processed with the BD rhapsody workflow (version 2.4b3.post1). All subsequent data analysis was performed in R (version 4.4.2) and Bioconductor (version 3.20) and tidyverse[101]. The V(D)J data (file BCR_VDJ_Dominant_Contigs_AIRR.tsv) was filtered for high quality, productive contigs and complete observation of heavy and light chain per cell (101,340 cells). The V(D)J data was combined with the mRNA Immune panel and sample hashtag data using the SingleCellExperiment package[102]. The default sample demultiplexing of the BD workflow was corrected using the emptyDrops function of the DropletUtils package[103]. This gave rise to a final set of 77835 cells that could reliably be assigned to a specific sample. The mRNA Immune panel expression data was used to classify B cells as memory or germinal center (GC) using clustering and gene marker lists[104]. The subsequent clonotype analysis included only B cells of IgG isotype (74015 cells). Amino acid substitution rates were calculated by aligning chains to the respective HkiL reference sequences using pairwiseAlignment and mismatchTable functions of the pwalign package[105]. Clonotypes were defined by 100% sequence identity of all six complement determining regions (CDRs) across heavy and light chain. Clonotype frequency distribution was calculated for each cluster and animal and the Gini-Simpson index was used to summarize the clonal diversity. To account for the difference in cell number per group, the index was averaged over 100 bootstraps based on the smallest group size (790 cells).

### Enzyme-linked immunosorbent assay (ELISA)

To determine the binding of monoclonal antibodies and of serum immunoglobulin to the GP-1 moiety of arenavirus glycoproteins we used the recombinant soluble GP1 domain of either LASV$_{LIV}$, LASV$_{LII}$ or LCMV, respectively, that were C-terminally fused to the human IgG1 Fc domain and expressed by transient transfection of expiCHO cells at the Protein Expression Core Facility, PECF, of the Swiss Federal Technical Highschool, EPFL, Lausanne, Switzerland[106]. For capture of these proteins on plates, goat anti-human IgG (Jackson Immunoresearch, 109-005-098) was diluted to 1 ng/µl in coating buffer (15 mM Na$_2$CO$_3$ and 35 mM NaHCO$_3$ in ddH$_2$O, pH: 9.6) and distributed in a 96-well high-affinity binding plates (Greiner Bio-one) and left overnight at 4 °C. Next day, the coating mix was flicked off, and 5% milk in PBS supplemented with 0.05% Tween-20 (PBST; Sigma, P9416) was added and left incubating for 1 h at RT to block non-specific binding. In separate

plates 3-fold serial dilutions of serum samples or the indicated antibody samples were prepared in PBST supplemented with 1% FBS, and subsequently the serially diluted samples were transferred into the coated and blocked ELISA plates. Samples were incubated for 1 h at 37 °C shaking at 300 rpm. Next, plates were washed 3 times with PBST and incubated with a secondary HRP-coupled antibody in PBST supplemented with 1% FBS for 1 h at RT at 300 rpm. Plates were washed 3 times with PBST and once with PBS prior to adding the color reaction mix (0.5 mg/ ml ABTS (Thermo Scientific, 34026), 28 mM citric acid, 44 mM (Na$_2$HPO$_4$) and 0.1% H$_2$O$_2$ in ddH$_2$O), which was incubated for 15–30 min. Color reaction was terminated by adding 1% of SDS (Sigma, 71729) in ddH$_2$O and measured optical density at a wavelength of 405 nm on an Infinite® M Plex device (TECAN). To determine the binding of monoclonal antibodies and of serum immunoglobulin to LASV$_{LIV}$-GPC and to LCMV-GPC we used a recombinant soluble form of these proteins (GPC-Streptag) consisting of the corresponding ectodomains fused to a C-terminal StreptagII.The ectodomain of the LCMV-GPC (WE strain) and of the LASV$_{LIV}$-GPC (Josiah strain) were C-terminally fused to the streptag II sequence and were expressed in transiently transfected expiCHO cells[15,66]. The ELISA assay was performed as above, except that GPC-Streptag was captured by coating ELISA plates with 2 µg/ml of Strep-TactinXT (Iba) instead of goat anti-human IgG. To quantify serum antibody responses and mAb binding, the area under the curve (AUC) of serum dilution factor or the EC$_{50}$ was determined, respectively. Naïve mouse serum and isotype control antibodies were used to determine the background signal of the assay, respectively.

### Monoclonal antibodies and virus neutralization assays

The monoclonal antibodies KL25 and WEN3 were originally elicited by infecting mice with LCMV[56]. The VSV-specific monoclonal antibody VI-7 served as isotype control.[77,107]. The LASV-GPC-specific antibodies 18.5 C and 12.1 F have been isolated from human LASV survivors[31]. For the present studies the published V(D)J sequences[33] were recombinantly expressed in conjunction with the mouse IgG2a constant domain. All recombinant antibodies in a mouse IgG2a or IgG1 format were produced by transient co-transfection of the respective heavy chain and light chain V(D)J expression plasmids in CHO cells (Protein Expression Core Facility, PECF, of the Swiss Federal Technical Highschool, EPFL, Lausanne, Switzerland). The antibodies were purified on an ÄKTAprime plus purification system using Protein G columns (GE healthcare). After 24 h of PBS dialysis, the purified antibodies were quantified by IgG ELISA. For the epitope binning ELISA with the objective of assessing the LASV$_{LIV}$ affinity matured KL25 IgG plates were subsequently incubated with KL25 IgM serum at 1:2000 dilution. To determine virus-neutralizing antibody titers of mouse serum

samples, the latter were heat-inactivated for 30 min at 56 °C. Serial dilutions of serum samples and monoclonal antibodies were prepared in MEM supplemented with 2% FBS. An equal volume of samples was incubated with a total of approximately 1000 IU (Infectious Units) replication-deficient rVSVΔG-EGFP pseudotyped with either LCMV-GPC (Genbank accession number AJ297484) or LASV$_{LIV}$-GPC (Genbank accession number NC_004296). Next, the serum – virus mixture was transferred into a 96-well flat-bottom plate and $3 \times 10^4$ VeroE6 cells per well were added. Plates were incubated at 37 °C for 16–18 h before fixing with 1% PFA for 10 min. The number of green (virus-infected) cells was quantified using an Immunospot S6 device (C.T.L.). Using a four-parameter nonlinear regression in GraphPad Prism the 50% neutralization titer (IC$_{50}$) was calculated based on the serum dilution factor that resulted in a half-maximal viral entry inhibition. The resulting values were reported as neutralizing titer.

### ADCC surrogate assay
The mFcγRIV ADCC Reporter Bioassay (Promega) was performed according to the manufacturer's instructions. Briefly, MDCK cells target cells stably expressing the LASV$_{LIV}$-GPC (kindly provided by Thomas Strecker, University of Marburg, Germany[108]) were seeded in white, flat-bottom 96-well plates (Corning) at a density of 20,000 cells per well in DMEM supplemented with 10% FBS and were incubated overnight at 37 °C in 5% CO$_2$. The following day, the medium was replaced with diluted mouse serum (1:100 in assay buffer), and 75,000 mFcγRIV-expressing Jurkat reporter cells were added per well. Negative control reference wells received assay buffer without serum. After incubation for 6 h at 37 °C in a 5% CO$_2$ atmosphere, 75 µl of luciferase assay substrate was added to each well and incubated for 10 min at room temperature. Luminescence was measured using a Tecan Spark plate reader (Tecan Group AG). Data were expressed as fold induction relative to the negative control reference wells (no serum control).

### Quantification and statistical analysis
For statistical testing we used GraphPad Prism software (Version 10.4.1, Graph Pad Software). Groups were tested for normal distribution before analysis using the Shapiro-Wilk normality test. Comparisons between two groups were performed using unpaired two-tailed Student's or Welch's t tests, when more than two groups were compared One-Way or Two-Way ANOVA was performed as appropriate. Dunnett's post-hoc test served to compare all groups against one reference group whereas Tukey's post-hoc test was conducted to compare all groups against each other. $p < 0.05$ was considered as statistically significant (*), $p < 0.01$ as highly significant (**) and $p \geq 0.05$ as not statistically significant. Cell counts and viral loads were log-converted for statistical analysis and the same was done for antibody AUC and IC$_{50}$ values when required to obtain a near-normal distribution.

### Reporting summary
Further information on research design is available in the Nature Portfolio Reporting Summary linked to this article.

## Data availability
The raw data generated in this study have been deposited in the Zenodo database under accession code https://doi.org/10.5281/zenodo.17794274 and are publicly available as of the date of publication of this article. Single cell RNA sequencing data have been deposited with the National Center for Biotechnology Information Gene Expression Omnibus (GEO) under the accession number GSE313953 and are publicly available as of the date of publication of this article. Source data are provided with this paper.

## Materials availability
Unique materials will be shared with qualified investigators under a material transfer agreement.

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

## Acknowledgements

This work was supported by the Swiss National Science Foundation (No. 310030_215043 to D.D.P.) and the EU-H2020-MSCA-COFUND EURIdoc programme (No.101034170 to D.D.P.). We wish to thank Thomas Strecker for providing LASV-GPC-expressing MDCK cells, Viktor Volchkov for VSVΔG/ZEBOVGP, Katrin Martin for helpful discussions, Karsten Stauffer for outstanding animal husbandry, Christian Beisel, Mirjam Feldkamp and the entire Genomics Facility Basel for scRNAseq, the entire DBM flow cytometry core facility for FACS-sorting, and the NIH Tetramer Core Facility for MHC tetramers. Bioinformatic analysis calculations were performed at sciCORE (http://scicore.unibas.ch/) scientific computing center at University of Basel.

## Author contributions

T.A-M., A.F.M., D.W., J.F., F.G., N.B., C.S., C.R., A.L.K, W.V.B., M.D., M.S., G.Z., M.P., and D.D.P. designed experiments. T.A-M., A.F.M., D.W., K.T., J.F., F.G., N.B., C.S., C.R., A.L.K., and M.L performed experiments. T.A-M., A.F.M., D.W., K.T., J.F., F.G., N.B., C.S., C.R., A.L.K., W.V.B, M.P., and D.D.P. analyzed data. T.A-M., A.F.M., and D.D.P. wrote the manuscript. D.W. and K.T. contributed equally to the present work.

## Competing interests

D.D.P. is a founder, consultant and shareholder of Hookipa Pharma Inc. commercializing arenavirus-based vector technology, and he as well as W.V.B. are listed as inventor on corresponding patents. The remaining authors declare no competing interests.
