## [Transparent Peer Review file · Nature Communications]

B cell immunity to the Lassa virus glycoprotein is a correlate of vaccination-induced virus control in mice

Corresponding Author: Professor Daniel Pinschewer

Version 0:

Reviewer comments:

Reviewer #1

(Remarks to the Author)

The authors claim to have identified “B-cell immunity to the Lassa virus glycoprotein as a correlate of vaccination-induced protection,” yet all supporting experiments are confined to an avirulent rLCMV/LASV surrogate that establishes transient viremia—not disease—in its natural rodent host. No survival, pathology, or immunosuppression end-points are measured, so the study never demonstrates protection from illness or lethality. In fact, the data merely show that pre-existing or vaccine-elicited antibodies can accelerate clearance of low-grade viremia in mice after an artificial challenge. Treating this outcome as a “correlate of protection” is therefore premature and, in my view, misleading.

The work’s potential impact on LASV vaccinology is further constrained by its heavy reliance on highly artificial systems. HkiL knock-in mice expressing a single, high-affinity LCMV GP1-specific B-cell receptor hardly represent the physiological human repertoire; yet major mechanistic conclusions (e.g. the necessity of AID-mediated hyper-mutation) rest on this model. Likewise, CD8-depleted C57BL/6J mice are immunologically permissive and—because *Mus musculus* is LCMV’s natural reservoir—are a notoriously poor predictor of LASV immunopathogenesis. Without corroboration in guinea-pigs, STAT-1-deficient mice, or non-human primates, the translational value of any “B-cell correlate” remains speculative at best.

Several claims conflict with, or simply ignore, established literature. The manuscript describes neutralising antibodies as “rarely elicited” and CD8 T cells as “first-order correlates,” yet cites only a single six-animal NHP study while omitting the growing list of reports that implicate CD4 T cells, monocytes and early innate responses (e.g. Baize et al. 2009; Mateo et al. 2021). More troublingly, the discussion of antibody-mediated protection omits multiple peer-reviewed demonstrations that non-neutralising antibodies do protect against bona-fide LASV challenge—including the recent eBioMedicine paper (DOI 10.1016/j.ebiom.2025.105647) that uses the very rVSVΔG-LASV-GPC vaccine evaluated here. This selective citation pattern exaggerates the novelty of the current study.

Data presentation invites further scepticism. Sample sizes are often three to five animals; groups are neither randomised nor blinded; and statistical reporting is inconsistent (e.g. no p-values for Fig. 2D, 2L, 5C, 5H). Figures 3C and 5B terminate viremia tracking at day 5 or day 14 without explanation, raising the possibility that transient control is followed by rebound—especially in the AID-mutant recipients where breakthrough viraemia is hinted but not explored. In Fig. 1C/D, one of the cross-reactive mAbs (WEN3) binds lineage IV GPC palpably weaker than the others; the text glosses over this discrepancy while still claiming broad cross-lineage reactivity.

Conceptual over-reach is apparent throughout. “Protection” is repeatedly used where the data support only “reduction in viremia”; “correlate” is used where merely an association is shown; and the Title and Abstract lines 29-30 ignore the study’s own admission (Methods, line 359) that rLCMV/LASV “is only an imperfect surrogate of LASV and does not induce haemorrhagic fever.” Until the authors demonstrate efficacy in a lethal model—or at least provide rigorous justification for equating viremia control with clinical benefit—the manuscript risks being cited misleadingly.

Minor editorial points

Lines 35-36: Most sources indicate at most 100-300k cases per year.

Candid#1 nomenclature (l. 47-49): capitalisation should be uniform.

Lines 61-63: add temporal context so non-LASV specialists appreciate treatment windows.

Line 70: citation 40 relates to humans—not guinea-pigs/NHPs—please adjust.

Line 71: capitalise the sentence start.

Line 88: specify “93 % amino-acid identity”.

Line 99: citation 44 does not support the statement—replace or delete.

Supplementary Fig. S4: panel labels mis-match the legend (“(C) is missing” at l. 808).

Fig. 6A legend (l. 878): specify “rVSV-eGFP control”.

Reminder: deposit all datasets on Zenodo and insert accession number before acceptance.

Reviewer #2

(Remarks to the Author)

Comments for Author :

This study provides compelling and well-executed evidence establishing B cell immunity to the Lassa virus glycoprotein (GPC) as a key correlate of vaccine-induced protection, independent of neutralizing antibodies or CD8+ T cells. The work addresses a critical gap in understanding LASV immunity, with significant implications for vaccine design and evaluation. The findings demonstrating cross-protective B cell responses induced by heterologous arenavirus glycoproteins, the requirement for affinity maturation shown by monoclonal B cell transfer, the protective role of non-neutralizing antibodies, and validation using a clinical-stage VSV-LASV vaccine are particularly noteworthy and original. While the manuscript is strong overall and suitable for publication in Nature Communications after addressing the points below, some aspects require clarification and additional discussion to fully solidify the conclusions.

Major Points:

1. The finding that non-neutralizing antibodies mediate protection is important and challenges prevailing views on LASV immunity. However, the manuscript would benefit from a more detailed discussion of potential Fc-mediated effector mechanisms (e.g., ADCC, antibody-dependent complement activation), which are likely to underlie the protective effects observed. If feasible, experimental validation using Fc receptor knockout models or in vitro effector function assays would substantially strengthen the conclusions.
2. The use of heterologous arenavirus glycoproteins to elicit cross-reactive protection is a notable strength. However, additional clarification of the breadth of this cross-reactivity—particularly across multiple LASV lineages or other arenaviruses—would increase the impact and generalizability of the findings.
3. The authors convincingly show that affinity maturation is essential for B cell-mediated protection using adoptive transfer models. Including quantitative data on somatic hypermutation or BCR repertoire evolution would enhance this point.

Reviewer #3

(Remarks to the Author)

Reviewer #4

(Remarks to the Author)

This is a nice study by Abreu-Mota et al that provides evidence of cross protective immunity across Lassa virus lineages and old-world arenaviruses by vaccination. Mechanistically B cell immunity against the LASV GP without the need for GP neutralization is identified as a correlate of protection. Overall, is a very elegant study, well conducted, and the conclusions are well justified by the data presented. I have a few questions that I think could help to clarify some of the main findings.

- 1) The authors demonstrate that when they use rLCMV expressing LASV GPC, viremia is achieved only when the cytoplasmic domain of LCMV GP is conserved. Why is the ‘acute’ non-viremic version rLCMV/LASVLIIA then utilized for immunization?
- 2) Does rLCMV/LASVLIIA replicate in order to generate enough antigen to trigger a T-cell dependent response? CD8 T cells are depleted to demonstrate T-cell independent protection but there is no evidence that a CD8 T cell response is even generated by immunization.
- 3) The authors demonstrate protection to challenge based on the presence of non-neutralizing Abs. Can they rule out off-target effects? Maybe challenging mice with a non-related virus? (e. g VSV, influenza).
- 4) What would be the biological basis for protection exerted by non-neutralizing antibodies? ADCC? Complement

mediated?

5) Viremia is essentially the only readout utilized in the paper but there are many examples of viremic mice that do not show disease signs. How about other morbidity markers such as weight loss, levels of serum aminotransferases, body scoring?

6) Please revise the manuscript to correct some typos and grammatical mistakes throughout

Version 1:

Reviewer comments:

Reviewer #1

(Remarks to the Author)

The authors have addressed most of our comments.

One big thing is that there still lacks a “future directions” section, where they propose similar work but in more straightforward Lassa challenge models (guinea pigs, STAT-1 mice, NHP) with bona fide Lassa.

Additional comments:

Comment 1.ii: I would specify that these clinical trials “...have identified viral load at admission to the healthcare facility as the primary predictor of LF outcome.”

Comment 4.3C: Regarding the Line 239-244, the authors mention that these experiments were stopped after 2 weeks because of a mutation in the GPC which corresponds to a known escape mutant of KL25. While there does not seem to be an associated Method section with this, the legend of Figure S4 indicates that the virus was grown from the blood of individual mice, and that the GPC was then sequenced. Because of this transient growing step, assuming in cell culture, what is the concern that this mutation was acquired from the in vitro passaging, rather than the in vivo challenge?

Reviewer #2

(Remarks to the Author)

Comments for the Author

The revised manuscript has fully and successfully addressed all concerns raised during the initial review process. The authors have performed a commendable job in implementing substantial revisions, including the refinement of terminology, a more nuanced discussion of model limitations, and the inclusion of critical new experimental data. These additions have significantly bolstered the manuscript's scientific rigor and clarity.

The updated data provide robust support for the study's central thesis: B cell immunity to the Lassa virus (LASV) glycoprotein (GPC) is a primary correlate of vaccine-induced protection, functioning independently of neutralizing antibodies or CD8⁺ T cells. This work offers high-impact insights into LASV immunology and vaccine development, meeting the rigorous standards for publication in Nature Communications.

Noteworthy Results:

1. Revelation of Cross-Reactive B Cell Immunity: Convincingly demonstrates that B cell immunity induced by the glycoproteins of distantly related arenaviruses (e.g., LCMV) can provide cross-protective control of LASV viral loads.
2. Establishment of Affinity Maturation as Critical: Through adoptive transfer experiments with monoclonal HkL B cells, provides direct evidence that activation of GPC-specific B cells and subsequent affinity maturation (via somatic hypermutation) are required for effective viremia control. The newly added single-cell V(D)J sequencing data (Fig. 5) offer quantitative support for this mechanism.
3. Challenge to Prevailing Paradigms: Clearly demonstrates the significant role of non-neutralizing antibodies in controlling viremia, challenging the traditional view that neutralizing activity is the sole or primary protective antibody function. The new ADCC surrogate assay data (Figs. S2C, S4C) provide supportive evidence for the potential involvement of effector functions like ADCC.
4. Validation with a Clinical-Stage Vaccine Candidate: Validation using a clinical-stage VSV-LASV vaccine candidate enhances the translational relevance and potential application of the findings. The new data (Fig. S6) further demonstrate the cross-lineage control elicited by this vaccine-induced immunity.

Significance and Originality:

This study represents a significant advancement in our understanding of protective immunity against LASV. By identifying B cell immunity—rather than T cells alone—as a central player under specific conditions, the authors fill a major gap in the field. The finding that heterologous glycoproteins can induce cross-protective B cell responses via affinity maturation is highly original and has direct implications for the design of “pan-arenavirus” vaccines. The manuscript is well-written and correctly contextualizes its findings within the current scientific landscape.

Overall Assessment:

The authors' response has been comprehensive and scientifically productive. The revised manuscript addresses all major questions, strengthens the evidence base, and presents conclusions with appropriate scientific nuance. The study provides novel, data-driven insights that are of broad interest to the fields of virology, immunology, and vaccinology. I recommend the

manuscript for publication in Nature Communications in its current form.

Reviewer #3

(Remarks to the Author)

Reviewer #4

(Remarks to the Author)

All my comments have been addressed. I thank the authors and have no further comments.

REVIEWER COMMENTS

Reviewer #1 (Remarks to the Author):

1. The authors claim to have identified “B-cell immunity to the Lassa virus glycoprotein as a correlate of vaccination-induced protection,” yet all supporting experiments are confined to an avirulent rLCMV/LASV surrogate that establishes transient viremia—not disease—in its natural rodent host. No survival, pathology, or immunosuppression end-points are measured, so the study never demonstrates protection from illness or lethality. In fact, the data merely show that pre-existing or vaccine-elicited antibodies can accelerate clearance of low-grade viremia in mice after an artificial challenge. Treating this outcome as a “correlate of protection” is therefore premature and, in my view, misleading.

We have addressed the reviewer’s critiques as follows:

i) We have reworked the entire manuscript including title and abstract to strictly avoid the term “protection” when referring to vaccination-induced suppression of rLCMV/LASV viremia in mice. Instead we use wording such as “viral load control”, “immune control” or “suppression of viremia”.

ii) Following the reviewer’s recommendation to “provide rigorous justification for equating viremia control with clinical benefit” (see below) we have included a paragraph into the manuscript’s introduction section outlining to the reader the strong evidence linking viral loads to clinical outcome in Lassa fever (Line 75-78): “[...]several large-scale clinical studies conducted in both children and adults, across different countries and over multiple decades, have identified viral load as the primary predictor of LF outcome¹⁻⁴, suggesting that viral-load suppression should be a central goal of vaccination-induced immunity.”

3. We have amended the manuscript’s text to put into context the viremia levels in rLCMV/LASV^C-infected mice (Line 121-123): “Mice infected with these “chronic variants” developed peak viremia of about 10^5 focus-forming units per milliliter of blood, comparable to LASV loads documented in humans, guinea pigs, and select NHP species during fatal LF⁴⁻⁸.”

2. The work’s potential impact on LASV vaccinology is further constrained by its heavy reliance on highly artificial systems. HkiL knock-in mice expressing a single, high-affinity LCMV GP1-specific B-cell receptor hardly represent the physiological human repertoire; yet major mechanistic conclusions (e.g. the necessity of AID-mediated hyper-mutation) rest on this model. Likewise, CD8-depleted C57BL/6J mice are immunologically permissive and—because *Mus musculus* is LCMV’s natural reservoir—are a notoriously poor predictor of LASV immunopathogenesis. Without corroboration in guinea-pigs, STAT-1-deficient mice, or non-human primates, the translational value of any “B-cell correlate” remains speculative at best.

In response to the reviewer’s critique we have amended the manuscript’s “Limitations of the study” paragraph as follows:

i) Line 443-445: “A further limitation of this work consists in the reliance on monoclonal HkiL B cells, which were useful for dissecting mechanistic aspects of the clonal evolution of LASV-GPC-specific antibody responses but may not fully represent the physiological human B-cell repertoire.”

ii) Line 431-435: “CD8 T cell depletion prevents both the clinical and biochemical manifestations of disease in mouse models of LF as well as in LCMV-infected mice⁹⁻¹⁴. While essential to distinguish between CD8 T cell- and antibody-mediated viral load control, CD8 T cell depletion therefore eliminated disease-related parameters as usable readouts in our study.”

Further in response to this reviewer’s critique and to a similar one by referee #4, we have included the new Fig. S2E showing that vaccination-induced CD8 T cells suppress rLCMV/LASV_{LIV}^C viremia in soluble antibody-deficient *slgM*^{-/-} *AID*^{-/-} mice. These data illustrate that the assessment of B cell-based suppression of viremia - the key aim of the present manuscript - required us to deplete CD8 T cells, which in return precluded a disease readout.

3. Several claims conflict with, or simply ignore, established literature. The manuscript describes neutralising antibodies as “rarely elicited” and CD8 T cells as “first-order correlates,” yet cites only a single six-animal NHP study while omitting the growing list of reports that implicate CD4 T cells, monocytes and early innate responses (e.g. Baize

et al. 2009; Mateo et al. 2021). More troublingly, the discussion of antibody-mediated protection omits multiple peer-reviewed demonstrations that non-neutralising antibodies do protect against bona-fide LASV challenge—including the recent eBioMedicine paper (DOI 10.1016/j.ebiom.2025.105647) that uses the very rVSVΔG-LASV-GPC vaccine evaluated here. This selective citation pattern exaggerates the novelty of the current study.

To address the reviewer's concerns we have amended the Introduction section to our manuscript as follows, aiming for a more balanced representation of the pertinent literature (Line 62-67): “[...],pre-clinical studies on a range of LF vaccine candidates have found that LASV-nAbs were inconsistently induced or reached only low titers^{8,15-23}, whereas robust T cell responses were commonly associated with vaccination-induced protection²⁰⁻²³. Moreover, studies comparing fatal and non-fatal LASV infection in unvaccinated non-human primates (NHPs) have observed that distinct innate immune response profiles as well as robust T cell responses predicted survival^{24,25}.”

Of note, however, Cooper and colleagues in the eBioMedicine paper cited by the reviewer do not draw clear-cut conclusions about the contribution of non-neutralizing antibodies to antiviral protection in their study: “These data suggest that antibodies developed by vaccinated macaques played a role in protection from LASV disease, but they do not exclude the possibility of significant contributions from T cells, as most of the vaccinated macaques did develop detectable peripheral blood T cells specific for GPC.”

4. Data presentation invites further scepticism. Sample sizes are often three to five animals; groups are neither randomised nor blinded; and statistical reporting is inconsistent (e.g. no p-values for Fig. 2D, 2L, 5C, 5H). Figures 3C and 5B terminate viremia tracking at day 5 or day 14 without explanation, raising the possibility that transient control is followed by rebound—especially in the AID-mutant recipients where breakthrough viraemia is hinted but not explored. In Fig. 1C/D, one of the cross-reactive mAbs (WEN3) binds lineage IV GPC palpably weaker than the others; the text glosses over this discrepancy while still claiming broad cross-lineage reactivity.

To dispel the reviewer's concerns we have taken the following measures:

1. We have re-worked our figure legends to point out more clearly that in cases where sample sizes are in the range of 5 animals, these data points represent only one out of two independent data sets.
2. Results of statistical analyses have been included in all figure panels for which differences are stated in the text.
3. We have amended the manuscript's text to outline the rationale for the day 5 time point in the experiment to Fig. 3C, and have further included the new figure panel S4A with its accompanying text explaining why viremia kinetics were ended around day 14 in the experiments to Figs. 4 and 5:

Line 206-207: “To determine direct antiviral effects we assessed viremia on day 5 after infection i.e. during the antibody's first half-life period²⁶ and prior to the onset of the CD8 T cell response²⁷.”

Line 239-244: “Of note, this experiment and similar ones below were ended at around two weeks after challenge since persisting low-level rLCMV/LASV_{LIV}^C viremia in a subset of HkiL+LASV_{LIV}^A animals was associated with mutations at amino acid 114 of the viral GPC, which corresponded to known KL25 escape mutations in LCMV-GPC²⁸ (Fig. S4A,B). Mutational escape would, therefore, have precluded a meaningful interpretation of the association between HkiL cell responses and viral load control at later time points.”

4. The text to Fig. 1C,D has been reworded as follows (Line 109-111): “Likewise, the mAbs KL25²⁹ and WEN3³⁰, which were isolated from LCMV-infected mice, exhibited LASV_{LIV}-GPC-specific reactivity, albeit WEN3 binding was comparably weak.”

5. Conceptual over-reach is apparent throughout. “Protection” is repeatedly used where the data support only “reduction in viremia”; “correlate” is used where merely an association is shown; and the Title and Abstract lines 29-30 ignore the study's own admission (Methods, line 359) that rLCMV/LASV “is only an imperfect surrogate of LASV and does not induce haemorrhagic fever.” Until the authors demonstrate efficacy in a lethal model—or at least provide rigorous justification for equating viremia control with clinical benefit—the manuscript risks being cited misleadingly.

As outlined in response to this reviewer's first point above, we have followed his/her recommendation to “provide rigorous justification for equating viremia control with clinical benefit” and have reworked the terminology throughout the entire manuscript.

Minor editorial points

6. Lines 35-36: Most sources indicate at most 100-300k cases per year.

To address the reviewer's question we have amended the respective sentence in the introduction section as follows (Line 44-46): "*While historically estimated to account for ~100'000 – 300'000 human infections each year³¹ more recent modeling of the virus' epidemiology projects 2.7 million annual LASV infections³².*"

Candid#1 nomenclature (l. 47-49): capitalisation should be uniform.

We have reworked the manuscript to consistently write "Candid#1".

Lines 61-63: add temporal context so non-LASV specialists appreciate treatment windows.

We have amended the respective sentence as follows (Line 73-75): "*Experimental work in guinea pigs and NHPs found that monoclonal antibody (mAb) therapy suppressed viremia within forty-eight hours and prevented disease even when therapeutically administered eight days after LASV exposure³³⁻³⁶.*"

Line 70: citation 40 relates to humans—not guinea-pigs/NHPs—please adjust.

The erroneous reference has been removed.

Line 71: capitalise the sentence start.

This deficiency has been corrected in the revised version of our manuscript.

Line 88: specify "93% amino-acid identity".

Following the reviewer's recommendation we have amended the sentence as follows (Line 103-104): "*Sequence conservation between the GPCs of different LASV lineages can be as low as ~93% amino acid identity (Fig. 1A,B).*"

Line 99: citation 44 does not support the statement—replace or delete.

Instead of the erroneous reference 44 (in the document provided first), which has been removed, this statement was meant to be supported by the publication by Lee et al. 2013, which has been added in the revised version of our manuscript.

Supplementary Fig. S4: panel labels mis-match the legend ("(C) is missing" at l. 808).

This deficiency has been corrected.

Fig. 6A legend (l. 878): specify "rVSV-eGFP control".

We have amended the figure legend to specify (Line 1194-1196): "*At wk-4, mice were immunized with either rVSVΔG-LASV-GPC, control rVSV vector (rVSV-EGFP) or they were left unvaccinated as indicated in the chart.*"

Reminder: deposit all datasets on Zenodo and insert accession number before acceptance.

We have amended the Data Availability section of our manuscript as follows (Line 722-726): "*Raw data of the experimental results reported in this study will be deposited with Zenodo and made publicly available as of the date of*

publication under the DOI 10.5281/zenodo.17794274. Single cell RNA sequencing data have been deposited with the National Center for Biotechnology Information Gene Expression Omnibus (GEO) under the accession number GSE313953.”

Reviewer #2 (Remarks to the Author):

Comments for Author :

This study provides compelling and well-executed evidence establishing B cell immunity to the Lassa virus glycoprotein (GPC) as a key correlate of vaccine-induced protection, independent of neutralizing antibodies or CD8+ T cells. The work addresses a critical gap in understanding LASV immunity, with significant implications for vaccine design and evaluation. The findings demonstrating cross-protective B cell responses induced by heterologous arenavirus glycoproteins, the requirement for affinity maturation shown by monoclonal B cell transfer, the protective role of non-neutralizing antibodies, and validation using a clinical-stage VSV-LASV vaccine are particularly noteworthy and original. While the manuscript is strong overall and suitable for publication in Nature Communications after addressing the points below, some aspects require clarification and additional discussion to fully solidify the conclusions.

Major Points:

1. The finding that non-neutralizing antibodies mediate protection is important and challenges prevailing views on LASV immunity. However, the manuscript would benefit from a more detailed discussion of potential Fc-mediated effector mechanisms (e.g., ADCC, antibody-dependent complement activation), which are likely to underlie the protective effects observed. If feasible, experimental validation using Fc receptor knockout models or in vitro effector function assays would substantially strengthen the conclusions.

Following this reviewer's recommendations and a related question from reviewer #4 we have conducted experiments that are shown in the new figure panels S2C and S4C and have amended the manuscript's text.

i) The new figure panels S2C and S4C are referred to as follows:

- Line 159-162: "[...] a surrogate assay for antibody-dependent cell-mediated cytotoxicity (ADCC) demonstrated that serum antibodies from rLCMV/LASV_{LIV}^A-immune mice triggered Fc gamma receptor IV- (FcγRIV-) mediated effector cell activation in the presence of target cells expressing LASV-GPC (Fig. S2C)."

- Line 251-253: "The ADCC surrogate assay revealed further that the serum of HkiL+LASV_{LIV}^A mice triggered substantially higher FcγRIV-mediated effector cell activation in the presence of LASV-GPC-expressing target cells than sera of the HkiL-only control group (Fig. S4C)."

ii) The revised discussion section states (line 389-393): "We found that LASV-GPC-specific secondary antibody responses, even when non-neutralizing, correlated with viral load control in CD8 T cell-depleted mice. Antibody effector functions such as ADCC (compare Figs. S2C, S4C) and antibody-dependent cell-mediated phagocytosis but also complement-mediated virion lysis may have contributed to this effect^{18,19,37}."

iii) We have included a section into the "Limitation of the study" paragraph, which explains (line 435-437): "Fcγ receptor-mediated effector functions are essential for antibody-mediated CD8 T cell depletion³⁸, precluding the use of Fcγ receptor-deficient mice to assess the Fcγ receptor-dependence of antibody-mediated viral load control in our experimental setting."

Of note in this context, the necessity to deplete CD8 T cells when aiming to read out B cell-mediated viral load control is underpinned by the new figure panels S2D,E (included in response to a specific request by reviewer #4), which are referred to in the text as follows (Line 168-174): "[...]rLCMV/LASV_{LIV}^A immunization elicited also viral epitope-specific CD8 T cell responses (Fig. S2D), and rLCMV/LASV_{LIV}^A-immunized slgM⁻xAID⁻ mice that were not depleted of CD8 T cells completely suppressed rLCMV/LASV_{LIV}^C challenge viremia (Fig. S2E). This observation indicated that not only specific antibodies but independently also vaccination-induced CD8 T cells were able to contain rLCMV/LASV_{LIV}^C challenge viremia, such that assessing the antiviral efficacy of the former required depletion of the latter."

2. The use of heterologous arenavirus glycoproteins to elicit cross-reactive protection is a notable strength. However, additional clarification of the breadth of this cross-reactivity—particularly across multiple LASV lineages or other arenaviruses—would increase the impact and generalizability of the findings.

To address the reviewer's question we have performed experiments that are included as new figure panels S6A-D, extending our findings to the glycoproteins of two additional LASV lineages. These data are referred to in the revised

manuscript text as follows (line 348-351): “*rVSVΔG-LASV-GPC-vaccinated and CD8 T cell-depleted mice suppressed not only the replication of rLCMV/LASV_{LIV}^C, which expresses the GPC contained in the vaccine, but they controlled viremia also when challenged with rLCMV/LASV_{LII}^C or rLCMV/LASV_{LIII}^C expressing the GPC of distinct LASV lineages (Fig. S6A-D).*”

3. The authors convincingly show that affinity maturation is essential for B cell-mediated protection using adoptive transfer models. Including quantitative data on somatic hypermutation or BCR repertoire evolution would enhance this point.

Following the reviewer’s recommendation we have performed single cell V(D)J repertoire sequencing (new Figs. 5 and S5). The data provide further support to the mechanistic postulate of our manuscript and are referred to in the revised text as follows (line 291-302): “*To compare the ability of different viral glycoproteins to promote HkiL cell receptor hypermutation and clonal diversification, we vaccinated TgL recipients of HkiL cells with either rLCMV/LASV_{LII}^A or rLCMV^A and five weeks later we processed progeny HkiL GC B cells and MBCs for single cell RNA sequencing-based V(D)J determination (Fig. 5A, S5A). At the time of analysis the repartition of HkiL progeny into GC B cells, MBC and ASCs was comparable irrespective of the vaccine administered (Fig. 5B). In notable contrast, the antibody light and heavy chain complementarity-determining regions (CDRs) of rLCMV/LASV_{LII}^A-induced HkiL GC B cells and also the light chain CDRs of MBCs from the same group of mice exhibited a significantly higher mutational burden than the respective sequences of rLCMV^A-activated HkiL cells (Fig. 5C-F). Accordingly, the clonal diversity of HkiL GC B cells was higher in rLCMV/LASV_{LII}^A- than in rLCMV^A-vaccinated animals as also evident in a higher Gini-Simpson diversity index, and an analogous trend was noted for HkiL MBCs (Fig. 5G,H). The observed CDR hypermutation exhibited a clear pattern with prominent recurrent amino acid exchanges (Fig. S5B,C).*”

Reviewer #3 (Remarks to the Author):

Reviewer #4 (Remarks to the Author):

This is a nice study by Abreu-Mota et al that provides evidence of cross protective immunity across Lassa virus lineages and old-world arenaviruses by vaccination. Mechanistically B cell immunity against the LASV GP without the need for GP neutralization is identified as a correlate of protection. Overall, is a very elegant study, well conducted, and the conclusions are well justified by the data presented. I have a few questions that I think could help to clarify some of the main findings.

1) The authors demonstrate that when they use rLCMV expressing LASV GPC, viremia is achieved only when the cytoplasmic domain of LCMV GP is conserved. Why is the 'acute' non-viremic version rLCMV/LASV_{LIIA} then utilized for immunization?

To address the reviewer's question we have amended the manuscript's text as follows (line 142-144):

"rLCMV/LASV_{LIIA} rather than its chronic counterpart rLCMV/LASV_{LIVC} was used for immunization in order to prime the immune system with LASV_{LIIA}-GPC without establishing viremic infection that would have interfered with the determination of rLCMV/LASV_{LIVC} viremia upon challenge."

2) Does rLCMV/LASV_{LIIA} replicate in order to generate enough antigen to trigger a T-cell dependent response? CD8 T cells are depleted to demonstrate T-cell independent protection but there is no evidence that a CD8 T cell response is even generated by immunization.

In response to the reviewer's question and a related comment by reviewer #1 we have included the new supplementary figure panels S2D,E. These results are referred to in the text as follows (line 168-174): [...] *"rLCMV/LASV_{LIIA} immunization elicited also viral epitope-specific CD8 T cell responses (Fig. S2D), and rLCMV/LASV_{LIIA}-immunized slgM^{-/-}xAID^{-/-} mice that were not depleted of CD8 T cells completely suppressed rLCMV/LASV_{LIVC} challenge viremia (Fig. S2E). This observation indicated that not only specific antibodies but independently also vaccination-induced CD8 T cells were able to contain rLCMV/LASV_{LIVC} challenge viremia, such that assessing the antiviral efficacy of the former required depletion of the latter."*

3) The authors demonstrate protection to challenge based on the presence of non-neutralizing Abs. Can they rule out off-target effects? Maybe challenging mice with a non-related virus? (e. g VSV, influenza).

In response to the reviewer's question we have amended the Results section as follows (line 145-156): *To control for the immunological specificity of vaccination-induced viral load suppression, a separate group of mice was immunized with an engineered LCMV (rLCMV/VSVG), which is molecularly identical to rLCMV/LASV_{LIIA} but expresses the antigenically unrelated vesicular stomatitis virus glycoprotein (VSVG) instead of LASV-GPC."* [...] *"Suppression of viremia by rLCMV/LASV_{LIIA}-immune but not by rLCMV/VSVG-vaccinated animals indicated that rLCMV/LASV_{LIVC} control relied largely on specific immunity to LASV-GPC."*

4) What would be the biological basis for protection exerted by non-neutralizing antibodies? ADCC? Complement mediated?

To address this reviewer's question and a related one from reviewer #2 we have conducted experiments that are shown in the new figure panels S2C and S4C and have amended the manuscript text:

i) The new figure panels S2C and S4C are referred to as follows:

- Line 159-162: “[...] a surrogate assay for antibody-dependent cell-mediated cytotoxicity (ADCC) demonstrated that serum antibodies from rLCMV/LASV_{LIV}^A-immune mice triggered Fc gamma receptor IV- (FcγRIV-) mediated effector cell activation in the presence of target cells expressing LASV-GPC (Fig. S2C).”

- Line 251-255: “The ADCC surrogate assay revealed further that the serum of HkiL+LASV_{LIV}^A mice triggered substantially higher FcγRIV-mediated effector cell activation in the presence of LASV-GPC-expressing target cells than sera of the HkiL-only control group (Fig. S4C). These findings suggested that antibodies produced by naïve HkiL cells did not neutralize LASV-GPC and failed to mediate substantial FcγR-mediated effector cell activation...”

ii) We have amended the discussion section (line 389-393): “We found that LASV-GPC-specific secondary antibody responses, even when non-neutralizing, correlated with viral load control in CD8 T cell-depleted mice. Antibody effector functions such as ADCC (compare Figs. S2C, S4C) and antibody-dependent cell-mediated phagocytosis but also complement-mediated virion lysis may have contributed to this effect^{18,19,37}.”

5) Viremia is essentially the only readout utilized in the paper but there are many examples of viremic mice that do not show disease signs. How about other morbidity markers such as weight loss, levels of serum aminotransferases, body scoring?

In response to this reviewer’s critique and a related comment by reviewer #1 we have amended the manuscript’s “Limitations of the study” paragraph as follows (line 431-435): “CD8 T cell depletion prevents both the clinical and biochemical manifestations of disease in mouse models of LF as well as in LCMV-infected mice⁹⁻¹⁴. While essential to distinguish between CD8 T cell- and antibody-mediated viral load control, CD8 T cell depletion therefore eliminated disease-related parameters as usable readouts in our study.”

Related to this, please see also our reply to this reviewer’s comment #2, referring to the new Fig. S2E, which shows that vaccination-induced CD8 T cells suppress rLCMV/LASV_{LIV}^C viremia in soluble antibody-deficient *slgM*^{-/-}*AID*^{-/-} mice. These data illustrate that the assessment of B cell-based suppression of viremia - the key aim of the present manuscript - required us to deplete CD8 T cells, which in return precluded a disease readout.

6) Please revise the manuscript to correct some typos and grammatical mistakes throughout

We have carefully reviewed the entire manuscript for typos and grammatical errors.

- 1 Strampe, J. *et al.* Factors associated with progression to death in patients with Lassa fever in Nigeria: an observational study. *Lancet Infect Dis* **21**, 876-886 (2021). [https://doi.org:10.1016/S1473-3099\(20\)30737-4](https://doi.org:10.1016/S1473-3099(20)30737-4)
- 2 Ogbaini-Emovon, E. *et al.* Virus Load Kinetics in Lassa Fever Patients Treated With Ribavirin: A Retrospective Cohort Study From Southern Nigeria. *Open Forum Infect Dis* **11**, ofae575 (2024). <https://doi.org:10.1093/ofid/ofae575>
- 3 Duvignaud, A. *et al.* Presentation and Outcomes of Lassa Fever in Children in Nigeria: A Prospective Cohort Study (LASCOPE). *J Pediatric Infect Dis Soc* **13**, 513-522 (2024). <https://doi.org:10.1093/jpids/piae083>
- 4 Johnson, K. M. *et al.* Clinical virology of Lassa fever in hospitalized patients. *J Infect Dis* **155**, 456-464 (1987).
- 5 Jahrling, P. B., Smith, S., Hesse, R. A. & Rhoderick, J. B. Pathogenesis of Lassa virus infection in guinea pigs. *Infect Immun* **37**, 771-778 (1982). <https://doi.org:10.1128/iai.37.2.771-778.1982>
- 6 Jahrling, P. B., Frame, J. D., Smith, S. B. & Monson, M. H. Endemic Lassa fever in Liberia. III. Characterization of Lassa virus isolates. *Trans R Soc Trop Med Hyg* **79**, 374-379 (1985). [https://doi.org:10.1016/0035-9203\(85\)90386-4](https://doi.org:10.1016/0035-9203(85)90386-4)
- 7 Peters, C. J. *et al.* Experimental studies of arenaviral hemorrhagic fevers. *Curr Top Microbiol Immunol* **134**, 5-68 (1987).
- 8 Brouwer, P. J. M. *et al.* Lassa virus glycoprotein nanoparticles elicit neutralizing antibody responses and protection. *Cell Host Microbe* **30**, 1759-1772 e1712 (2022). <https://doi.org:10.1016/j.chom.2022.10.018>
- 9 Flatz, L. *et al.* T cell-dependence of Lassa fever pathogenesis. *PLoS Pathog* **6**, e1000836 (2010). <https://doi.org:10.1371/journal.ppat.1000836>

- 10 Oestereich, L. *et al.* Chimeric Mice with Competent Hematopoietic Immunity Reproduce Key Features of
Severe Lassa Fever. *PLoS Pathog* **12**, e1005656 (2016). <https://doi.org:10.1371/journal.ppat.1005656>
- 11 Maruyama, J. *et al.* CD4 T-cell depletion prevents Lassa fever associated hearing loss in the mouse model.
PLoS Pathog **18**, e1010557 (2022). <https://doi.org:10.1371/journal.ppat.1010557>
- 12 Straub, T. & Pircher, H. Enhancing immunity prevents virus-induced T-cell-mediated immunopathology in B
cell-deficient mice. *Eur J Immunol* **49**, 782-789 (2019). <https://doi.org:10.1002/eji.201847962>
- 13 Leist, T. P., Ruedi, E. & Zinkernagel, R. M. Virus-triggered immune suppression in mice caused by virus-
specific cytotoxic T cells. *J Exp Med* **167**, 1749-1754 (1988). <https://doi.org:10.1084/jem.167.5.1749>
- 14 Zinkernagel, R. M. *et al.* T cell-mediated hepatitis in mice infected with lymphocytic choriomeningitis virus.
Liver cell destruction by H-2 class I-restricted virus-specific cytotoxic T cells as a physiological correlate of
the 51Cr-release assay? *J Exp Med* **164**, 1075-1092 (1986). <https://doi.org:10.1084/jem.164.4.1075>
- 15 Warner, B. M., Safronetz, D. & Stein, D. R. Current perspectives on vaccines and therapeutics for Lassa
Fever. *Viol J* **21**, 320 (2024). <https://doi.org:10.1186/s12985-024-02585-7>
- 16 Enriquez, A. S. *et al.* Mapping the antibody response to Lassa virus vaccination of non-human primates.
EBioMedicine **114**, 105673 (2025). <https://doi.org:10.1016/j.ebiom.2025.105673>
- 17 Cooper, C. L. *et al.* Preclinical development of a replication-competent vesicular stomatitis virus-based
Lassa virus vaccine candidate advanced into human clinical trials. *EBioMedicine* **114**, 105647 (2025).
<https://doi.org:10.1016/j.ebiom.2025.105647>
- 18 Abreu-Mota, T. *et al.* Non-neutralizing antibodies elicited by recombinant Lassa-Rabies vaccine are critical
for protection against Lassa fever. *Nat Commun* **9**, 4223 (2018). <https://doi.org:10.1038/s41467-018-06741-w>
- 19 Ronk, A. J. *et al.* A Lassa virus mRNA vaccine confers protection but does not require neutralizing antibody
in a guinea pig model of infection. *Nat Commun* **14**, 5603 (2023). <https://doi.org:10.1038/s41467-023-41376-6>
- 20 Geisbert, T. W. *et al.* Development of a new vaccine for the prevention of Lassa fever. *PLoS Med* **2**, e183
(2005). <https://doi.org:10.1371/journal.pmed.0020183>
- 21 Mateo, M. *et al.* A single-shot Lassa vaccine induces long-term immunity and protects cynomolgus monkeys
against heterologous strains. *Sci Transl Med* **13** (2021). <https://doi.org:10.1126/scitranslmed.abf6348>
- 22 Carnec, X. *et al.* A Vaccine Platform against Arenaviruses Based on a Recombinant Hyperattenuated
Mopeia Virus Expressing Heterologous Glycoproteins. *J Virol* **92** (2018). <https://doi.org:10.1128/JVI.02230-17>
- 23 Fisher-Hoch, S. P., Hutwagner, L., Brown, B. & McCormick, J. B. Effective vaccine for lassa fever. *J Virol* **74**,
6777-6783 (2000).
- 24 Baize, S. *et al.* Early and strong immune responses are associated with control of viral replication and
recovery in lassa virus-infected cynomolgus monkeys. *J Virol* **83**, 5890-5903 (2009).
<https://doi.org:10.1128/JVI.01948-08>
- 25 Baillet, N. *et al.* Systemic viral spreading and defective host responses are associated with fatal Lassa fever
in macaques. *Commun Biol* **4**, 27 (2021). <https://doi.org:10.1038/s42003-020-01543-7>
- 26 Vieira, P. & Rajewsky, K. The half-lives of serum immunoglobulins in adult mice. *Eur J Immunol* **18**, 313-316
(1988). <https://doi.org:10.1002/eji.1830180221>
- 27 Berghaler, A. *et al.* Envelope exchange for the generation of live-attenuated arenavirus vaccines. *PLoS
Pathog* **2**, e51 (2006). <https://doi.org:10.1371/journal.ppat.0020051>
- 28 Hangartner, L., Zinkernagel, R. M. & Hengartner, H. Antiviral antibody responses: the two extremes of a
wide spectrum. *Nat Rev Immunol* **6**, 231-243 (2006). <https://doi.org:10.1038/nri1783>
- 29 Bruns, M., Cihak, J., Muller, G. & Lehmann-Grube, F. Lymphocytic choriomeningitis virus. VI. Isolation of a
glycoprotein mediating neutralization. *Virology* **130**, 247-251 (1983).
- 30 Seiler, P. *et al.* Induction of protective cytotoxic T cell responses in the presence of high titers of virus-
neutralizing antibodies: implications for passive and active immunization. *J Exp Med* **187**, 649-654 (1998).
<https://doi.org:10.1084/jem.187.4.649>
- 31 McCormick, J. B., Webb, P. A., Krebs, J. W., Johnson, K. M. & Smith, E. S. A prospective study of the
epidemiology and ecology of Lassa fever. *J Infect Dis* **155**, 437-444 (1987).

- 32 Smith, D. R. M. *et al.* Health and economic impacts of Lassa vaccination campaigns in West Africa. *Nat Med* **30**, 3568-3577 (2024). <https://doi.org:10.1038/s41591-024-03232-y>
- 33 Cross, R. W. *et al.* A human monoclonal antibody combination rescues nonhuman primates from advanced disease caused by the major lineages of Lassa virus. *Proc Natl Acad Sci U S A* **120**, e2304876120 (2023). <https://doi.org:10.1073/pnas.2304876120>
- 34 Cross, R. W. *et al.* Antibody therapy for Lassa fever. *Curr Opin Virol* **37**, 97-104 (2019). <https://doi.org:10.1016/j.coviro.2019.07.003>
- 35 Li, H. *et al.* A cocktail of protective antibodies subverts the dense glycan shield of Lassa virus. *Sci Transl Med* **14**, eabq0991 (2022). <https://doi.org:10.1126/scitranslmed.abq0991>
- 36 Mire, C. E. *et al.* Human-monoclonal-antibody therapy protects nonhuman primates against advanced Lassa fever. *Nat Med* **23**, 1146-1149 (2017). <https://doi.org:10.1038/nm.4396>
- 37 Evgin, L. *et al.* Complement inhibition enables tumor delivery of LCMV glycoprotein pseudotyped viruses in the presence of antiviral antibodies. *Mol Ther Oncolytics* **3**, 16027 (2016). <https://doi.org:10.1038/mto.2016.27>
- 38 White, J. M. *et al.* Removal of Fc Glycans from [(89)Zr]Zr-DFO-Anti-CD8 Prevents Peripheral Depletion of CD8(+) T Cells. *Mol Pharm* **17**, 2099-2108 (2020). <https://doi.org:10.1021/acs.molpharmaceut.0c00270>

Reviewer #1 (Remarks to the Author):

The authors have addressed most of our comments.

One big thing is that there still lacks a “future directions” section, where they propose similar work but in more straightforward Lassa challenge models (guinea pigs, STAT-1 mice, NHP) with bona fide Lassa.

Following the reviewer’s recommendation we have amended the Discussion section as follows (line 427): “It will therefore be important that the key concepts derived from this work be independently confirmed and corroborated in commonly used LASV challenge models.”

Additional comments:

Comment 1.ii: I would specify that these clinical trials “...have identified viral load at admission to the healthcare facility as the primary predictor of LF outcome.”

In response to the reviewer’s comment we have amended the respective sentence as follows (line 73): “[...] several large-scale clinical studies conducted in both children and adults, across different countries and over multiple decades, have identified viral load, measured either on hospital admission or during the course of illness, as the primary predictor of LF outcome.” While references 38-40 have only monitored viral loads at hospital admission, reference 41 (Johnson et al. JID 1987) reports differential viremia throughout the course of illness.

Comment 4.3C: Regarding the Line 239-244, the authors mention that these experiments were stopped after 2 weeks because of a mutation in the GPC which corresponds to a known escape mutant of KL25. While there does not seem to be an associated Method section with this, the legend of Figure S4 indicates that the virus was grown from the blood of individual mice, and that the GPC was then sequenced. Because of this transient growing step, assuming in cell culture, what is the concern that this mutation was acquired from the in vitro passaging, rather than the in vivo challenge?

To address the reviewer’s concern we have amended the Results section as follows (line 242): “The absence of such mutations from viruses circulating in LASV_{L11}^A-only mice indicated these mutations were the result of HkIL cell-driven *in vivo* selection, ...”.

Reviewer #2 (Remarks to the Author):

Comments for the Author

The revised manuscript has fully and successfully addressed all concerns raised during the initial review process. The authors have performed a commendable job in implementing substantial revisions, including the refinement of terminology, a more nuanced discussion of model limitations, and the inclusion of critical new experimental data. These additions have significantly bolstered the manuscript’s scientific rigor and clarity.

The updated data provide robust support for the study’s central thesis: B cell immunity to the Lassa virus (LASV) glycoprotein (GPC) is a primary correlate of vaccine-induced protection, functioning independently of neutralizing antibodies or CD8⁺ T cells. This work offers high-impact insights into LASV immunology and vaccine development, meeting the rigorous standards for publication in Nature Communications.

Noteworthy Results:

- 1. Revelation of Cross-Reactive B Cell Immunity: Convincingly demonstrates that B cell immunity induced by the glycoproteins of distantly related arenaviruses (e.g., LCMV) can provide cross-protective control of LASV viral loads.*
- 2. Establishment of Affinity Maturation as Critical: Through adoptive transfer experiments with monoclonal HkIL B cells, provides direct evidence that activation of GPC-specific B cells and subsequent affinity maturation (via somatic hypermutation) are required for effective viremia control. The newly added single-cell V(D)J sequencing data (Fig. 5) offer quantitative support for this mechanism.*
- 3. Challenge to Prevailing Paradigms: Clearly demonstrates the significant role of non-neutralizing antibodies in controlling viremia, challenging the traditional view that neutralizing activity is the sole or primary protective antibody function. The new ADCC surrogate assay data (Figs. S2C, S4C) provide supportive evidence for the potential involvement of effector functions like ADCC.*
- 4. Validation with a Clinical-Stage Vaccine Candidate: Validation using a clinical-stage VSV-LASV vaccine candidate enhances the translational relevance and potential application of the findings. The new data (Fig. S6) further demonstrate the cross-lineage control elicited by this vaccine-induced immunity.*

Significance and Originality:

This study represents a significant advancement in our understanding of protective immunity against LASV. By identifying B cell immunity—rather than T cells alone—as a central player under specific conditions, the authors fill a major gap in the field. The finding that heterologous glycoproteins can induce cross-protective B cell

responses via affinity maturation is highly original and has direct implications for the design of "pan-arenavirus" vaccines. The manuscript is well-written and correctly contextualizes its findings within the current scientific landscape.

Overall Assessment:

The authors' response has been comprehensive and scientifically productive. The revised manuscript addresses all major questions, strengthens the evidence base, and presents conclusions with appropriate scientific nuance. The study provides novel, data-driven insights that are of broad interest to the fields of virology, immunology, and vaccinology. I recommend the manuscript for publication in Nature Communications in its current form.

Reviewer #3 (Remarks to the Author):

Reviewer #4 (Remarks to the Author):

All my comments have been addressed. I thank the authors and have no further comments.